# SE(3)-Equivariant Attention Networks for Shape Reconstruction in Function Space

**Evangelos Chatzipantazis**[*]**, Stefanos Pertigkiozoglou**[*]
University of Pennsylvania
{vaghat,pstefano}@seas.upenn.edu

**Edgar Dobriban**
University of Pennsylvania
dobriban@wharton.upenn.edu

**Kostas Daniilidis**
University of Pennsylvania
kostas@cis.upenn.edu

## Abstract

We propose a method for 3D shape reconstruction from unoriented point clouds. Our method consists of a novel SE(3)-equivariant coordinate-based network (TF-ONet), that parametrizes the occupancy field of the shape and respects the inherent symmetries of the problem. In contrast to previous shape reconstruction methods that align the input to a regular grid, we operate directly on the irregular point cloud. Our architecture leverages equivariant attention layers that operate on local tokens. This mechanism enables local shape modelling, a crucial property for scalability to large scenes. Given an unoriented, sparse, noisy point cloud as input, we produce equivariant features for each point. These serve as keys and values for the subsequent equivariant cross-attention blocks that parametrize the occupancy field. By querying an arbitrary point in space, we predict its occupancy score. We show that our method outperforms previous SO(3)-equivariant methods, as well as non-equivariant methods trained on SO(3)-augmented datasets. More importantly, local modelling together with SE(3)-equivariance create an ideal setting for SE(3) scene reconstruction. We show that by training only on single, aligned objects and without any pre-segmentation, we can reconstruct novel scenes containing arbitrarily many objects in random poses without any performance loss.

## 1 Introduction

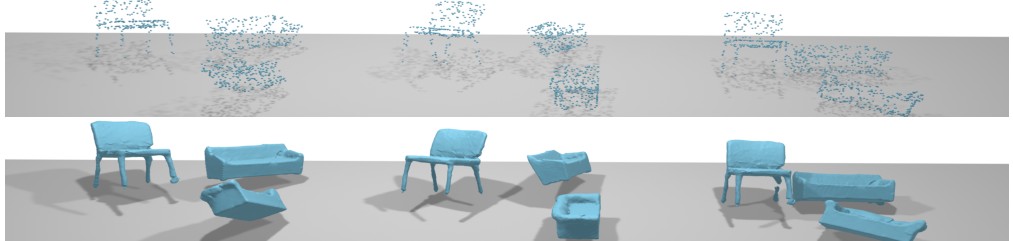

Figure 1: (Above): A scene-level point cloud produced by individual SE(3)-transformations of three sparse object point clouds. (Below): Our equivariant reconstruction. *The whole scene* is given as input to our network. The network, trained only on *single objects in canonical poses* and agnostic to the number, position and orientation of the objects is able to reconstruct the scene accurately.

With the advent of range sensors in robotics and in medical applications, research in shape reconstruction from point clouds has seen an increasing activity (Berger et al., 2017). The performance of classical optimization methods tends to degrade when point clouds become sparser, noisier, unoriented, or untextured. Deep learning methods have been proven useful in encoding shape priors, and solving the reconstruction problem end to end (Riegler et al., 2017). Many of these deep learning methods operate on meshes (Wang & Zhang, 2022; Gong et al., 2019), voxels (Riegler et al., 2017), and point clouds (Qi et al., 2016). While voxels are easy to manipulate, shape resolution is limited by memory. On the other hand, meshes can guarantee watertight reconstructions, but

---

[*]Equal Contribution

they only handle a predefined topology. Point clouds are lightweight in terms of memory, but they discard topology. Recently proposed deep learning methods represent the geometry via a learned occupancy map, or a signed distance function (SDF). In particular, the seminal works of Mescheder et al. (2019); Park et al. (2019) inspired many follow-up works (Chen & Zhang, 2019b; Genova et al., 2020; Sitzmann et al., 2019). Such representations can encode arbitrary topologies with an effectively infinite resolution.

According to Kendall, "Shape is the geometry of an object modulo position, orientation, and scale" (Kendall, 1989). While intensive research in the field (Niemeyer & Geiger, 2021; Peng et al., 2020; Niemeyer et al., 2020) has led to increasingly better results, very few of these methods incorporate symmetries as an inductive bias for learning. Most translation-equivariant reconstruction methods build on the convolutional occupancy network (Peng et al., 2020), while most SO(3)-equivariant architectures (Zhu et al., 2021), and their extensions to SE(3) with GraphOnet (Chen et al., 2022), use the equivariant modules from Vector Neurons (Deng et al., 2021). We propose TF-Onet, a novel SE(3)-equivariant coordinate-based network for shape reconstruction. Motivated by the SE(3)-transformer (Fuchs et al., 2020), we design a two-level network that uses equivariant attention modules. The first level, acting as an encoder, extracts local features from the point cloud by applying self-attention in local neighborhoods around each point. The second level, a cross-attention occupancy network, takes as input the extracted point features and the coordinates of a query point in space, and outputs the value of the occupancy function at the specified query point.

Even unique objects consist of smaller primitive parts, whose subsets are subsequently composed to form large collections of objects. This property extends naturally to scenes that are created by a composition of objects. Our method performs local shape modeling by leveraging the expressivity of equivariant local attention modules and generalizes to novel scenes with novel configurations of objects from classes unseen during training. This property distinguishes our method from similar equivariant works that either use global features (Deng et al., 2021) or per-point features that encode long-range dependencies by using subsampling to expand their receptive field (Chen et al., 2022). Additionally, as we describe in Section 3.3 the use of the Tensor Field framework allows our method to utilize higher order representations in contrast to the previous works which use Vector Neurons and thus are constrained to only use type-0 (scalars) and type-1 (vectors) representations. In Section 4, we provide experimental evidence showcasing how these differences benefit our method in the reconstruction of single objects in arbitrary poses and in the reconstruction of novel scenes.

**Our contributions can be summarized as follows**:
- We propose TF-Onet, a novel SE(3)-equivariant, coordinate-based, attention network for learning occupancy fields from sparse point clouds and use it for surface reconstruction.
- Experimentally, we outperform other equivariant coordinate-based networks (Vector Neurons, GraphOnet) and non-equivariant networks (Occupancy Networks, Convolutional Occupancy Networks, IFNet, NeuralPull) trained with augmentations.
- The most compelling property of our method is that equivariance and local shape modeling allows our network to produce high-quality reconstructions of novel scenes while being trained only on single-aligned objects. These scenes contain an arbitrary number of objects in random poses. We show quantitative 5a and qualitative 6a performance gap over previous methods in a synthetic dataset of randomly placed objects (*Seismic dataset*). We also show qualitative results on the more challenging Matterport3D (Chang et al., 2017) containing real scenes with unseen object classes.

## 2   RELATED WORK

In this section, we discuss previous work on surface reconstruction from input point clouds. We focus on methods that reconstruct the surface of an object by using either an occupancy function or a SDF. For oriented point clouds (with known normal vectors), the occupancy function or the SDF can be constructed by classical methods that do not require learning (Alexa et al., 2003; Kazhdan & Hoppe, 2013). These methods tend to fail in the presence of noise, or when the input point cloud is sparse. To surpass such limitations, Mescheder et al. (2019); Chen & Zhang (2019a) proposed to learn the occupancy function for each input point cloud. Similarly, Park et al. (2019) proposed to learn to infer the SDF of the object's surface from a sparse set of SDF values. A limitation of the above methods is that they use a global feature vector—or code—to represent the whole object (or scene), which limits their ability to generalize to novel scenes or objects. More recent methods

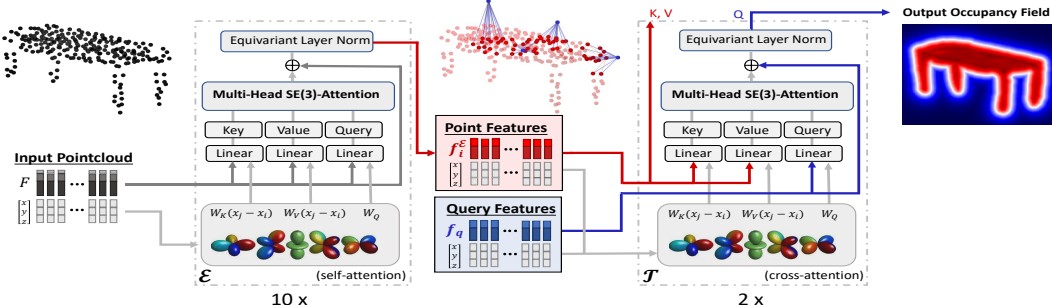

Figure 2: Our method consists of two networks $\mathcal{E}$, $\mathcal{T}$. First, to each point $\vec{x}_i$ in the point cloud, we assign a simple local feature $f_i$ of type-1 (rotating as a vector) which can be the relative position of the point $\vec{x}_i$ to the centroid of its local neighborhood. Similarly, to each query point $\vec{q} \in \mathbb{R}^3$ we assign a local feature $f_q$. Given a point cloud consisting of pairs $(\vec{x}_i, f_i)$, the network $\mathcal{E}$ applies SE(3)-equivariant attention to assign to each point $\vec{x}_i$ a learned feature $f_i^{\mathcal{E}}$. Finally, given the learned feature-augmented point cloud and the query-feature pair $(\vec{q}, f_q)$, the network $\mathcal{T}$ applies SE(3)-equivariant cross-attention to output the scalar value of the occupancy function at point $\vec{q}$.

leverage the similarities between local patches of different objects by learning either a combination of local and global features (Genova et al., 2020; Erler et al., 2020), or only local ones (Jiang et al., 2020; Chabra et al., 2020; Tretschk et al., 2020; Boulch & Marlet, 2022; Williams et al., 2022; Chen et al., 2022). The composition of these local features allows the reconstruction of scenes containing a variety of different objects. To define the local neighborhoods for which the local features are extracted, Jiang et al. (2020); Chabra et al. (2020); Tretschk et al. (2020) voxelize the space into a regular grid and learn a feature representation for each voxel, while Boulch & Marlet (2022); Chen et al. (2022) learns a local feature representation for each input point. In our work, we follow the latter approach by using local attention layers (Bahdanau et al., 2015) that dynamically change the attention weights of each point to its neighbors. Attention layers were popularized with the introduction of the transformer architecture (Vaswani et al., 2017), and were later applied in computer vision tasks such as image classification (Dosovitskiy et al., 2021; Wu et al., 2021). Specifically for point cloud processing tasks, Pan et al. (2021) used both local and global attention for object detection, and Yu et al. (2021) used a transformer for point cloud completion.

There is a large body of work on incorporating known symmetries into the learning process, tracing back to Fukushima (1980); LeCun et al. (1989) with the use of convolutional layers to build translation-equivariant networks. More recent works extend this idea to discrete (Cohen & Welling, 2016) and continuous (Weiler et al., 2018b) rotationally invariant architectures. These methods have also been applied beyond Euclidean domains, to spheres (Esteves et al., 2018), graphs (Maron et al., 2019; Brandstetter et al., 2022), meshes (de Haan et al., 2021), and general manifolds (Cohen et al., 2019; Weiler et al., 2021). For transformer architectures, Fuchs et al. (2020) propose an SE(3)-equivariant transformer by using the results of Tensor Field Networks (Thomas et al., 2018). Romero & Cordonnier (2021) proposed a framework for constructing linear self-attention layers that are equivariant to arbitrary discrete groups. Operating on point clouds, an SE(3)-equivariant network (Chen et al., 2021) performs pose estimation and classification using SE(3) convolutions. For the problem of surface reconstruction from point clouds, Convolutional Occupancy Networks (Peng et al., 2020) and later methods (Lionar et al., 2021; Tang et al., 2021; Boulch & Marlet, 2022) use convolutional layers to achieve translation-equivariance. Finally, Vector Neurons (Deng et al., 2021) propose an architecture for SO(3)-equivariant reconstruction from point clouds, extended by GraphOnet Chen et al. (2022) for the SE(3)-equivariant case.

## 3 METHOD

### 3.1 LEARNING THE OCCUPANCY FIELD

Consider a 3D point cloud $P = \{(\vec{x}_i, f_i)\}_{i=1}^N$, where $\vec{x}_i \in \mathbb{R}^3$ denotes the spatial location of a point and $f_i \in \mathbb{R}^M$ is an (optional) associated feature. If the features include the normal vectors, the point cloud is called *oriented*. Otherwise it is called *unoriented*. We denote the matrix of coordinates by $X = [\vec{x}_1, \cdots, \vec{x}_N] \in \mathbb{R}^{3 \times N}$ and the matrix of features by $F = [f_1, \cdots, f_N] \in \mathbb{R}^{M \times N}$.

Given a point cloud, our goal is to infer the shape from which it was sampled. We represent this shape with an *occupancy function* $o^* : \mathbb{R}^3 \to \{0, 1\}$ whose 1-level set, $\{\vec{x} \in \mathbb{R}^3 | o^*(\vec{x}) = 1\}$,

encodes the volume occupied by the object and whose boundary encodes the surface of the object. We approximate this function by learning an operator $\mathcal{T}$ that acts on an input point cloud and produces a function $\hat{o}_P : \mathbb{R}^3 \to [0, 1]$ that describes the occupancy score $\hat{o}_P(\vec{q})$ of each position $\vec{q} \in \mathbb{R}^3$. Following Mescheder et al. (2019), we parametrize $\mathcal{T}$ by a coordinate-based neural network. In other words, given a point cloud $P = (X, F)$, and an arbitrary point $\vec{q}$ in space, we predict $\mathcal{T}[X, F](\vec{q}) = \hat{o}_P(\vec{q}) \in [0, 1]$. In contrast to Mescheder et al. (2019), our architecture works directly on the irregular point grid, extracting a local signature to describe the point cloud, and constraining the model to achieve SE(3)-equivariance, thus respecting the symmetries of the problem.

It is more common, especially in robotics applications, to receive featureless point clouds (for example, from Lidar sensors). Our paper focuses on *sparse, noisy, and featureless (in particular, unoriented)* point clouds. Since the performance of $\mathcal{T}$ depends on the expressivity of the input features $F$, we first design a feature extractor (or encoder) $\mathcal{E}$ that produces a point cloud along with features, $\mathcal{E}[X] = (X, F)$. Overall, $\hat{o}_P(\vec{q}) = [\mathcal{T}(\mathcal{E}(X; \phi))](\vec{q}; \theta)$. Our design of $\mathcal{T}, \mathcal{E}$ is founded on two basic principles: *local shape modelling* and *equivariance*. Objects can be seen as a composition of local parts and local surfaces. At this level, even dissimilar objects may share structure. Local shape modelling leverages this compositionality to learn from multiple shapes in a more effective way. This is particularly helpful for reconstructing novel object classes.

On the other hand, SE(3)-equivariance restricts models to produce consistent occupancy maps irrespective of the rigid transformations of the shape. This can further reduce sample complexity. Together, these two properties create an ideal setting for scene reconstruction. Scenes are composed of objects which appear in different orientations and positions. Without leveraging compositionality and equivariance, every new configuration of objects would be seen as a novel scene by the network, and thus learning would be hindered by the combinatorial explosion of these combinations. We discuss how we impose the above properties on $\mathcal{T}, \mathcal{E}$ in the next sections.

We note two more important properties of point cloud processing. First, there is no canonical ordering on the points, so a simultaneous permutation of the columns of $X, F$ describes the same point cloud (*permutation equivariance*) and second, the number $N$ can vary between point clouds. Thus the input has a set structure, irregularly embedded in a Euclidean domain. Two main architectures that deal with such inputs are Point Cloud Convolutions (Wu et al., 2019) and Attention modules (Yu et al., 2021). We opt for the latter in our design because point cloud convolutions, when constrained to be SE(3)-equivariant, result in very few free parameters (Thomas et al., 2018) and in particular they eliminate the angular degrees of freedom from learning. Thus the performance is highly dependent on ad-hoc non-linearities (Poulenard & Guibas, 2021). Attention modules on the other hand are inherently non-linear and the attention kernel can be viewed as a modulation to the angular profile of the weights, thus having more degrees of freedom (Fuchs et al., 2020).

### 3.2 USING ATTENTION FOR LOCAL SHAPE MODELLING

**Preliminaries on Attention:** We propose to construct our networks $\mathcal{E}, \mathcal{T}$ as a composition of equivariant attention modules. Given $N_{\text{out}}$ query tokens $Q_i \in \mathbb{R}^{d_Q}$ and $N_{\text{in}}$ key-value pairs $K \in \mathbb{R}^{d_Q \times N_{\text{in}}}, V \in \mathbb{R}^{d_V \times N_{\text{in}}}$, an attention module can be described by the formula:

$$A(Q_i, K, V) = V \operatorname{softmax}(K^T Q_i) = \sum_{j=1}^{N_{\text{in}}} \underbrace{\frac{\exp\left(Q_i^T K_j\right)}{\sum_{j'=1}^{N_{\text{in}}} \exp\left(Q_i^T K_{j'}\right)}}_{\alpha(Q_i, K_j)} V_j, \; i \in [N_{\text{out}}].$$

where subscripts index columns. For each query token $Q_i$, the output $A$ is a linear combination of the values $V_j$, modulated by the similarity $\alpha(Q_i, K_j)$ of the query $Q_i$ to the corresponding key $K_j$, as imposed by the attention kernel $\alpha$. In *self-attention* layers the queries, keys and values are functions of the same set of features, while in *cross-attention* the query features are distinct and only the keys and values are functions of the same features. We design the feature extractor $\mathcal{E}$ as a composition of $L$ self-attention layers, and the occupancy operator $\mathcal{T}$ as a composition of $L'$ cross-attention layers.

**Self-attention feature extractor $\mathcal{E}$:** We associate each point with a query token with the same functional dependency

$$Q = Q(\vec{x}_i, f) = W_Q f.$$

Moreover, we design the self-attention module so that not all queries have access to all the keys, but a query attends only keys depending on its position. In particular, if $\vec{x}_j \in \mathcal{N}(\vec{x}_i)$, with $\mathcal{N}(\vec{x}_i)$ the

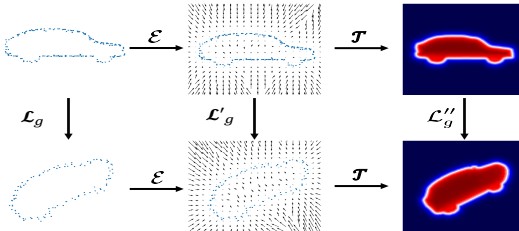

Figure 3: (Up): Input point cloud, input query field (type-1), output occupancy field (type-0) and (Down) their roto-translations. Here $\mathcal{L}'_g$ describes the action on both the query field $(\vec{q}, S'(X, q))$ depicted above and the point cloud field $(X, F_{\mathcal{E}})$ (described in the main text); $\mathcal{T}, \mathcal{E}$ are equivariant which is equivalent to the diagram being commutative, i.e., $\mathcal{L}''_g \circ \mathcal{T} = \mathcal{T} \circ \mathcal{L}'_g$ and $\mathcal{E} \circ \mathcal{L}_g = \mathcal{L}'_g \circ \mathcal{E}$.

$k$ nearest neighbors of $\vec{x}_i$, we assign to the pair $(\vec{x}_i, \vec{x}_j)$ a key-value token. The functional form of keys and values is

$$V(\vec{x}_j, \vec{x}_i, f) = W_V(\vec{x}_j - \vec{x}_i)f, \quad K(\vec{x}_j, \vec{x}_i, f) = W_K(\vec{x}_j - \vec{x}_i)f.$$

The standard way to incorporate relative positional encoding is via concatenation or addition (Vaswani et al., 2017). Our functional form $W_K(\vec{x}_j - \vec{x}_i)f$ can be understood as a more general way to encode the positions of the tokens. This generalization will be important in our case when we impose extra constraints to satisfy rotational equivariance.

The self-attention module of the encoder $\mathcal{E}$ at layer $l \geq 1$ gets a feature-augmented point cloud $(X, F)$ and produces a new feature-augmented pointcloud $(X, F')$ where $F'$ is computed as:

$$\mathcal{E}^l[X, F]_i = \sum_{\vec{x}_j \in \mathcal{N}(\vec{x}_i)} \frac{\exp\left[(W_Q^l f_i)^T (W_K^l(\vec{x}_j - \vec{x}_i)f_j)\right]}{\sum_{\vec{x}_j \in \mathcal{N}(\vec{x}_i)} \exp\left[(W_Q^l f_i)^T (W_K^l(\vec{x}_j - \vec{x}_i)f_j)\right]} W_V^l(\vec{x}_j - \vec{x}_i)f_j, \quad (1)$$

Since the input point cloud $X$ includes only the locations of the points without additional features, we design a function $\tilde{\mathcal{E}}^0$ that associates hand-designed (not learned) features to the points. In this work, $\tilde{\mathcal{E}}^0[X] = S(X)$ where $S(X)_i = \vec{x}_i - 1/|\mathcal{N}(x_i)| \sum_{x_j \in \mathcal{N}(x_i)} \vec{x}_j$ is the relative position from the neighborhood's centroid. Then, $\mathcal{E} = \mathcal{E}^L \circ \cdots \circ \mathcal{E}^1 \circ \tilde{\mathcal{E}}^0$.

**Cross-attention occupancy network $\mathcal{T}$:** We design the occupancy operator $\mathcal{T}$ as a composition of $L'$ cross-attention layers. The input query to $\mathcal{T}$ can be any point $\vec{q} \in \mathbb{R}^3$ to which we assign a token $Q(\vec{q}, f_q) = W'_Q f_q$. The output of $\mathcal{T}$ is the occupancy value of that query location. The keys and values in $\mathcal{T}$ are constructed from the output of the feature extractor $\mathcal{E}[X] = (F^{\mathcal{E}} := [f_1^{\mathcal{E}}, ..., f_N^{\mathcal{E}}])$. In particular, if $\vec{x}_j \in \mathcal{N}_X(\vec{q})$, we create a key-value pair for $(\vec{q}, \vec{x}_j)$ with the form $K(\vec{x}_j, \vec{q}, f_j^{\mathcal{E}}) = W'_K(\vec{x}_j - \vec{q})f_j^{\mathcal{E}}$ and $V(\vec{x}_j, \vec{q}, f_j^{\mathcal{E}}) = W'_V(\vec{x}_j - \vec{q})f_j^{\mathcal{E}}$. Given a feature-augmented point $q^l := (\vec{q}, f_q^l)$, the cross-attention module $\mathcal{T}^l$ outputs a new feature-augmented point $q^{l+1} = (\vec{q}, f_q^{l+1})$. These new features are computed as:

$$\mathcal{T}^l[X, F^{\mathcal{E}}](q^l) = \sum_{\vec{x}_j \in \mathcal{N}_X(\vec{q})} \frac{\exp\left[(W_Q'^l f_q^l)^T (W_K'^l(\vec{x}_j - \vec{q})f_j^{\mathcal{E}}\right]}{\sum_{\vec{x}_j \in \mathcal{N}_X(\vec{q})} \exp\left[(W_Q'^l f_q^l)^T (W_K'^l(\vec{x}_j - \vec{q})f_j^{\mathcal{E}})\right]} W_V'^l(\vec{x}_j - \vec{q})f_j^{\mathcal{E}}, \quad (2)$$

Since the input query includes only its location $\vec{q}$ and not additional features, first we associate it with a fixed (not learned) feature $f_q^1 := S'(X, q) = \vec{q} - \frac{1}{|\mathcal{N}_X(\vec{q})|} \sum_{\vec{x}_j \in \mathcal{N}_X(\vec{q})} \vec{x}_j$, where $\mathcal{N}_X(\vec{q}) = \mathcal{N}(\arg\min_{\vec{x}_i \in X} \|\vec{q} - \vec{x}_i\|_2)$. Then after passing the feature augmented point to the remaining layers of $\mathcal{T}$ we get the output $(\vec{q}, f_q^{L'+1})$ which corresponds to the occupancy value at point $\vec{q}$. Thus the self-attention and cross-attention modules process the features on the point cloud and the 3d field respectively, without changing the topology and by performing local attention.

## 3.3 EQUIVARIANT ATTENTION FOR SHAPE RECONSTRUCTION

Shape reconstruction should be independent of the coordinate system used, including of the position of the origin and the orientation of the coordinate axes. One way to capture this geometric consistency is via SE(3)-equivariance, which can be formulated in the language of group theory. In our setup, neither $\mathcal{E}$ nor $\mathcal{T}$ satisfy this property without additional constraints on $(W_Q, W_K, W_V)$ and $(W'_Q, W'_K, W'_V)$. In this section we formulate and solve these constraints. We will assume familiarity with equivariance, and defer a more extensive discussion to the Appendix.

**Equivariance constraints:** In our formulation, the equivariance constraint is that the occupancy field should be a *type-0 (or scalar) field* i.e., a simultaneous roto-translation of the point cloud and the query should result in an invariant prediction. Formally, for all $(T, R) \in \mathrm{SE}(3), \vec{q} \in \mathbb{R}^3$,

$$\mathcal{T}[\mathcal{E}(RX + \oplus_N T)](R\vec{q} + T) = \mathcal{T}[\mathcal{E}(X)](\vec{q}). \tag{3}$$

This constraint has to be satisfied from input to output, not at every layer. For intermediate layers we can relax the constraint and output more expressive vector or tensor fields, with pre-specified transformation properties, so that the whole composition of layers results in a scalar field.

**Per-layer equivariance constraints.** We can think of a feature-augmented point cloud $(X, F)$ as a feature field in $\mathbb{R}^3$ with finite support. Then, both $\mathcal{E}, \mathcal{T}$ process fields in $\mathbb{R}^3$ at each layer. We need to specify how these fields transform under a roto-translation of $X$. Different transformation laws in the layers correspond to different constraints on their weights. We design the layers to transform as,

$$\mathcal{E}^l(RX + \oplus_N T, \rho_{\mathcal{E}}^l(R)F^l) = \rho_{\mathcal{E}}^{l+1}(R)F^{l+1} \tag{4}$$

$$\mathcal{T}^l[\mathcal{E}(RX + \oplus_N T)](R\vec{q} + T, \rho_{\mathcal{T}}^l(R)f_q^l) = \rho_{\mathcal{T}}^{l+1}(R)f_q^{l+1} \tag{5}$$

for all $(T, R) \in SE(3)$, where $\rho_{\mathcal{E}}^l(R), \rho_{\mathcal{T}}^l(R)$ are *SO(3)-representations* i.e., square invertible matrices satisfying $\rho(R_1 R_2) = \rho(R_1)\rho(R_2)$, for all $R_1, R_2 \in SO(3)$. The layers $\tilde{\mathcal{E}}^0, \tilde{\mathcal{T}}^0$ have also been designed to admit this transformation law since:

$$\tilde{\mathcal{E}}^0(RX + \oplus_N T) = R\tilde{\mathcal{E}}^0(X)$$

$$\tilde{\mathcal{T}}^0[\mathcal{E}(RX + \oplus_N T)](R\vec{q} + T) = R\tilde{\mathcal{T}}^0[\mathcal{E}(X)](\vec{q})$$

i.e., $\rho_{\mathcal{E}}^1(R) = \rho_{\mathcal{T}}^1(R) = R$ (proof Appendix A.10). These per-layer constraints in Eqs. 4, 5 are sufficient to solve Eq. 3, provided that $\rho_{\mathcal{T}}^{L'+1}(R) = I$. These transformation laws have the form of the *induced representation of SE(3) via SO(3)* (Fulton & Harris, 2013) (Appendix A.8.4).

**Per-layer constraints on the weights**: Observe that the transformation laws above describe fields whose features stay invariant under a translation of their domains (i.e., the point cloud and $\mathbb{R}^3$) but when this domain rotates they are multiplied by a square matrix $\rho(R)$. Such fields are called $\rho$-fields. It is clear from Eq. 1, 2 that due to the use of relative positional encoding, each feature is translation invariant; and thus each field is translation equivariant. We focus on rotations next, discussing constraints on $\mathcal{E}$, and adapting them to $\mathcal{T}$. We prove in the Appendix that to solve Eq. 4, it suffices to satisfy for all $R \in SO(3)$ and $\vec{x}_i, \vec{x}_j, j \neq i$ the following constraints on the weights:

$$\begin{cases} W_Q^l \rho_{\mathcal{E}}^l(R) = \rho_{\mathcal{E}}^{l+1}(R)W_Q^l \\ W_K^l(R(\vec{x}_j - \vec{x}_i))\rho_{\mathcal{E}}^l(R) = \rho_{\mathcal{E}}^{l+1}(R)W_K^l(\vec{x}_j - \vec{x}_i) \\ W_V^l(R(\vec{x}_j - \vec{x}_i))\rho_{\mathcal{E}}^l(R) = \rho_{\mathcal{E}}^{l+1}(R)W_V^l(\vec{x}_j - \vec{x}_i). \end{cases} \tag{6}$$

This reduces the per-layer constraints in Eq. 4 to constraints on the weights of each layer. By solving these constraints, we uncover the free parameters in each layer.

**Feature types and irreducibles:** To solve equation 6 it is necessary to exploit the structure of the matrices $\rho_{\mathcal{E}}^l, \rho_{\mathcal{T}}^l$, which is an object of study of representation theory (Appendix A.8). Specifically, according to Peter-Weyl theorem we can block diagonalize a representation $\rho : SO(3) \to \mathbb{R}^{d \times d}$ as

$$\rho(R) = Q^T[\oplus_k D_k(R)]Q$$

where $Q$ is a change of basis matrix and $D_k : SO(3) \to \mathbb{R}^{(2k+1) \times (2k+1)}, k \in \mathbb{N}$ is the $k$-th irreducible representation—i.e., non-decomposable into smaller one—of SO(3). Those $D_k(R)$ matrices produced by the decomposition are called $k$-th Wigner-D matrices. If a field decomposes to a single irreducible $D_k$, i.e., $\rho = D_k$ for some $k \in \mathbb{N}$, it is called a *type-k* field. We already discussed a type-0 (or scalar) field $\rho(R) = D_0(R) = I$, the occupancy field, as well as a type-1 (or vector) field $\rho(R) = D_1(R) = R$ on the input point cloud $(X, S(X))$. In practice our feature fields $f$ on each point are composed of multiple copies (or *multiplicities*) of irreducibles of different types that form complex $\rho-$fields i.e., $f = \oplus_{k,m_l} f_{k,m_l}$, where $k$ indexes the type and $m_k$ its multiplicity.

**Layer Parametrization**: Given $f^l = \bigoplus_{k \in \mathcal{K}} \bigoplus_{m_k} f_{k,m_k}^l$, and $f^{l+1} = \bigoplus_{k' \in \mathcal{K}'} \bigoplus_{m_{k'}} f_{k',m_{k'}}^{l+1}$, we denote by $W^{k'k}$ a block of the matrix $W$ that maps a $k$-type to a $k'$-type (there are many such blocks depending on the multiplicities). For clarity, we drop the layer index $l$ from the matrices, since all

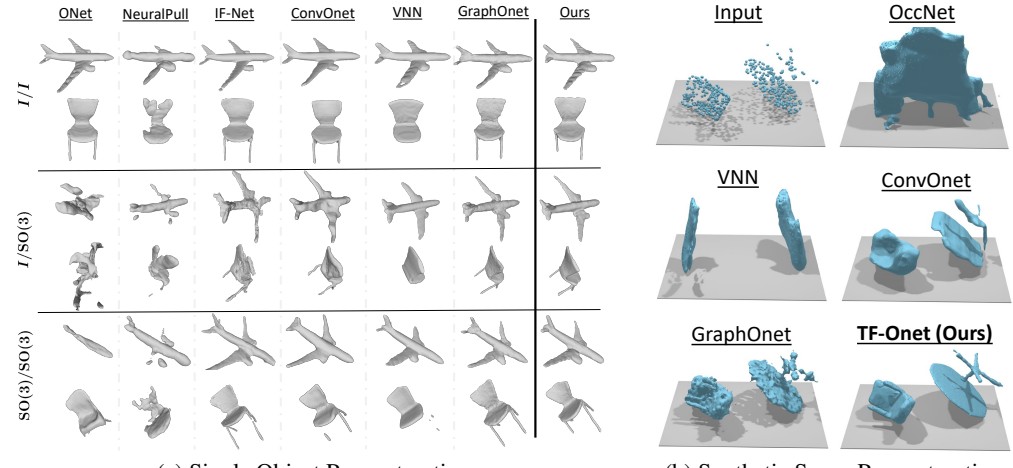

(a) Single Object Reconstruction    (b) Synthetic Scene Reconstruction

Figure 4: (a) Qualitative results for single object surface reconstruction for models trained and evaluated in three modes: training and testing on aligned shapes ($I/I$), training on aligned shapes and testing on rotated ones ($I/\mathrm{SO}(3)$), training and testing on rotated shapes ($\mathrm{SO}(3)/\mathrm{SO}(3)$). (b) Scene reconstructions from the *Seismic* dataset, using models trained on aligned single objects.

constraints are similar. By requiring that the attention kernel is a scalar field of the positions, we find in Appendix A.10 the sufficient conditions :

$$\left\{\begin{array}{l} W_Q^{k'k} D_k(R) = D_{k'}(R) W_Q^{k'k} \\ W_K^{k'k}(R(\vec{x}_j - \vec{x}_i)) D_k(R) = D_{k'}(R) W_K^{k'k}(\vec{x}_j - \vec{x}_i) \\ W_V^{k'k}(R(\vec{x}_j - \vec{x}_i)) D_k(R) = D_{k'}(R) W_V^{k'k}(\vec{x}_j - \vec{x}_i). \end{array}\right\} \tag{7}$$

The solution of the first equation follows from Schur's Lemma (Appendix A.8.1). The next two have been studied in Weiler et al. (2018a); Thomas et al. (2018). Finally, the inner product in the attention kernel can zero out some of the parameters in the keys:

$$\left\{\begin{array}{l} W_Q^{kk} = w_Q^{kk} I_{2k+1}, W_Q^{k'k} = 0, k \neq k' \\ W_V^{k'k}(\vec{x}_j - \vec{x}_i) = \sum_{J=|k'-k|}^{k'+k} \phi_{J,V}^{k'k}(\|\vec{x}_j - \vec{x}_i\|; \theta_V) C_J^{k'k}(\frac{\vec{x}_j - \vec{x}_i}{\|\vec{x}_j - \vec{x}_i\|}) \\ W_K^{k'k}(\vec{x}_j - \vec{x}_i) = \sum_{J=|k'-k|}^{k'+k} \phi_{J,K}^{k'k}(\|\vec{x}_j - \vec{x}_i\|; \theta_K) C_J^{k'k}(\frac{\vec{x}_j - \vec{x}_i}{\|\vec{x}_j - \vec{x}_i\|}), \ W_K^{k'k} = 0, k' \notin \mathcal{K} \end{array}\right. \tag{8}$$

where $C_J^{k'k}(\hat{x}) = \sum_{m=-J}^{J} Y_{Jm}(\hat{x}) Q_{Jm}^{k'k}$, $\hat{x} = \vec{x}/\|\vec{x}\|$ and $Q_{Jm}^{k'k} \in \mathbb{R}^{(2k'+1) \times (2k+1)}$ are slices from the Clebsch-Gordan matrices $Q^{k'k}$, $Y^J : S^2 \to \mathbb{R}^{2J+1}$ is the $j$-th real-valued spherical harmonic and $Y_{Jm}(\hat{x}) = [Y_J(\hat{x})]_m$ is its $m$-th coordinate. The remaining free parameters are the, $\phi_J$-s which we parametrize with small MLPs. (See Appendix A.7 for an example of these solutions)

The constraints for the cross-attention module $\mathcal{T}$ are similar to Eq. 7. However, we can further restrict to type-1 features for the keys in the first layer, i.e., $[W_K]^{i,j} = 0$, for $i \neq 1$, without any loss. The reason is that the input query field is of type-1. In Fig. 3 we visualize the equivariance constraint on $\mathcal{T}, \mathcal{E}$ by using a commutative diagram. For additional expressivity, we perform equivariant multi-headed attention by splitting the channels, i.e., the multiplicities of the irreducible representations. We also use skip connections and equivariant layer normalization as in Thomas et al. (2018), but defer the details to the Appendix. See Fig. 2 for an overview of the method.

**Irreducible types for shape reconstruction:** In deep networks, the difference between stacked channels and those forming a $\rho$-field is that the latter are equipped with transformation laws to mix channels, providing a geometric meaning to the representations. We select the irreducibles to describe our intuition about the geometric entities we look for. If we want to learn a feature that behaves like a 3D Euclidean vector under rotation (for example a normal vector) we need to construct a type-1 feature. If we want a feature that behaves like a symmetric matrix under rotations (such as the inertial matrix), we need a 6-dimensional channel that comprises of one type-0 (scalar) channel and five channels that compose a type-2 (traceless symmetric matrix) feature. We incorporate such geometric entities in a distinct way from previous SO(3)-based methods in surface reconstruction, which only handle vector fields (Deng et al., 2021; Zhu et al., 2021; Chen et al., 2022). In particular, type-2 features can be useful in our problem to define the bending of the local surface in the receptive field of the query. Thus they are useful to infer whether the query is inside the shape. Our intuitions are supported by the experimental performance gain compared to Deng et al. (2021); Chen et al. (2022) and the ablation study on the types of representations used in our network, shown in Table 1.

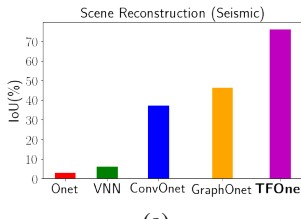 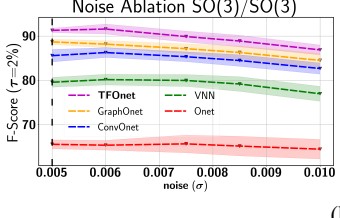 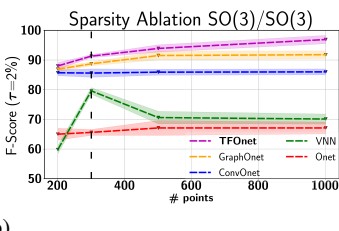

(a)  (b)

Figure 5: (a) Quantitative results of reconstruction on the *Seismic* dataset (synthetic scenes) for models trained on single aligned objects. (b) Ablation study on the effect of point cloud density and noise. All models are trained on 300 points with added normal noise of standard deviation 0.005 (black dashed line in the figure) and are evaluated on different sparsity and noise settings.

Table 1: Chamfer-L1 distance, F-Score and IoU achieved by different methods on single object reconstruction from sparse point clouds (300 points) sampled from ShapeNet. We evaluate our method (TF-Onet) on three different versions: (E:0-1,D:0-1) the encoder and the decoder use up to type-1 representations, (E:0-2,D:0-1) the encoder uses up to type-2 and the decoder uses up to type-1 representations, (E:0-2,D:0-2) the encoder and the decoder use up to type-2 representations

| | CHAMFER-L1 $\downarrow$ | | | F-SCORE ($\tau = 1\%$) $\uparrow$ | | | F-SCORE ($\tau = 2\%$) $\uparrow$ | | | IoU $\uparrow$ | | |
|---|---|---|---|---|---|---|---|---|---|---|---|---|
| | $I/I$ | $I$/SO(3) | SO(3)/SO(3) | $I/I$ | $I$/SO(3) | SO(3)/SO(3) | $I/I$ | $I$/SO(3) | SO(3)/SO(3) | $I/I$ | $I$/SO(3) | SO(3)/SO(3) |
| ONET | 0.1 | 0.4 | 0.2 | 66.4% | 21.4% | 39.3% | 89.7% | 40.4% | 65.6% | 77.8% | 30.9% | 58.2% |
| CONVONET | **0.093** | 0.16 | 0.12 | **71.7%** | 43.9% | 58.9% | **92.0%** | 73.8% | 85.6% | 77.2% | 57.8% | 71.2% |
| VNN | 0.14 | 0.14 | 0.15 | 56.5% | 56.5% | 56.8% | 81.2% | 81.2% | 79.6% | 69.3% | 69.3% | 68.8% |
| IF-NET | 0.095 | 0.17 | 0.13 | 68% | 43.5% | 55.9% | 91.7% | 73.5% | 84.5% | 77.3% | 49.4% | 69.9% |
| NEURALPULL | 0.18 | 0.17 | 0.17 | 50.4% | 50.5% | 50.5% | 73.2% | 73.6% | 73.6% | 64.8% | 65.7% | 65.7% |
| GRAPHONET | 0.105 | 0.105 | 0.104 | 67.1% | 67.1% | 67.2% | 88.7% | 88.7% | 88.7% | 73.2% | 73.2% | 73.2% |
| TFONET(E:0-1,D:0-1) | 0.105 | 0.105 | 0.105 | 66% | 66% | 66.1% | 88.9% | 88.9% | 88.9% | 73.8% | 73.8% | 73.9% |
| TFONET(E:0-2,D:0-1) | 0.095 | 0.095 | 0.096 | 69.2% | 69.2% | 69.2% | 90.9% | 90.9% | 90.4% | 77.4% | 77.4% | 77.3% |
| TFONET(E:0-2,D:0-2) | **0.093** | **0.093** | **0.093** | 71.2% | **71.2%** | **71.1%** | 91.3% | **91.3%** | **91.3%** | **78%** | **78%** | **78%** |

## 4 EXPERIMENTS

We perform experiments with surface reconstruction from unoriented sparse and noisy input point clouds. We show the importance of SE(3)-equivariance by evaluating objects in various poses and positions in space. Additionally, we show how local shape modeling and equivariance allows our method to *train only on single aligned objects* and generalize to *scenes containing multiple objects in arbitrary locations and orientations*.

### 4.1 SINGLE OBJECT RECONSTRUCTION FROM A SPARSE POINT CLOUD

In our first experiment, we train and evaluate our model on sparse point clouds sampled from single objects in ShapeNet (Chang et al., 2015). For each input shape, we first uniformly sample 300 points from the ground truth mesh, and then we add normal noise with zero mean and standard deviation of 0.005. We note that this experimental setting of 300 points is more challenging compared to previous works that used denser point clouds containing 3000 points.

First we train and evaluate on the original objects from ShapeNet, where both the training and test data points are in their canonical position (the $I/I$ case). Additionally, we follow Deng et al. (2021) and evaluate on test data points transformed by random SO(3) rotations. We evaluate models that were trained either on aligned training data (the $I$/SO(3) case), or on training data augmented by SO(3) rotations (the SO(3)/SO(3) case). In the case of our method (TF-Onet), as shown in Table 1, we experiment with different choices for the type of the intermediate representations used by our encoder and decoder. By adjusting the number of channels we make sure that all of our models have the same number of learnable parameters. We compare with the Occupancy Network (Onet) (Mescheder et al., 2019)—a non-equivariant network that extracts a global feature; with the Convolutional Occupancy Network (ConvOnet) (Peng et al., 2020) and the Implicit Feature Network (IF-Net) (Chibane et al., 2020)—two translation equivariant networks that extract local features; with Neural Pull (Baorui et al., 2021)—a method that focuses on reconstruction from dense point clouds by performing only test time optimization; with Vector Neurons (VNN) (Deng et al., 2021), —a SO(3) equivariant network that extracts a global feature; and with GraphOnet (Chen et al., 2022), — a SE(3) equivariant network that subsamples the input point cloud to extract per point features with long range dependencies. We note that in contrast to our method both VNN and GraphOnet can only use up to type-1 equivariant features.

In Table 1, we present the F-Score (Tatarchenko et al., 2019), the Chamfer-L1 distance (Fan et al., 2017) and the Intersection over Union (IoU) of the reconstruction, for models trained on the aligned and SO(3)-augmented datasets. We refer to section A.1 of the Appendix for a more detailed de-

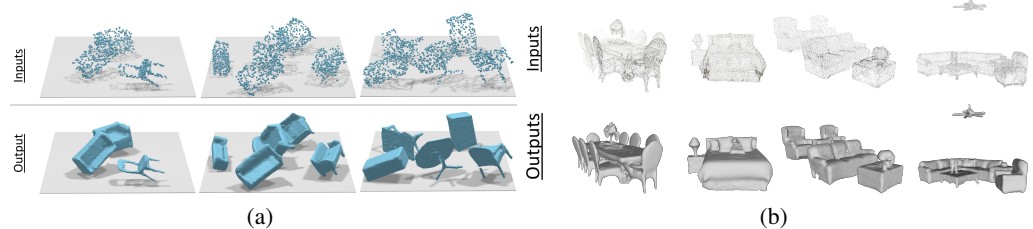

Figure 6: Examples of scene reconstructions using our method, trained only on aligned single objects from ShapeNet. (a) Reconstruction of synthetic scenes from the seismic dataset, (b) Reconstructions of realistic scenes from Matterport3D. (Chang et al., 2017)

scription of the evaluation metrics. Our method achieves consistent performance regardless of the rotation of the training or testing data points. Additionally, the best performance is achieved when we use up to type-2 representations in both the encoder and the decoder (E:0-2,D:0-2). When the testing data are transformed by random $SO(3)$ rotations, our method consistently outperforms the compared methods, regardless of whether it is trained on rotated or aligned training data. In figure 4a, we show qualitative comparisons between the methods. Our model achieves consistently high quality reconstruction in all settings ($I/I$, $I/SO(3)$, $SO(3)/SO(3)$). Finally, to study how the point cloud density and the added noise affects the reconstruction, we train in the original setting and evaluate on point clouds with different densities and levels of noise. As shown in figure 5b, our method outperforms previous methods on sparser point clouds, and scales better to denser point clouds.

## 4.2 SCENE RECONSTRUCTION WITH SINGLE OBJECT TRAINING

In this section, we evaluate the ability of our model to reconstruct novel scenes with many different objects, while *trained only on single objects*. Due to SE(3)-equivariance, performance is consistent and independent of the pose and the position of the objects. Additionally, our method performs computations in local neighborhoods that usually contain points from a single object. These two factors allow us to reconstruct scenes that contain objects in arbitrary poses and positions, without the need to train on similar scenes, or to segment into separate objects and reconstruct each one. We construct a dataset of synthetic rooms with multiple objects from ShapeNet in arbitrary locations and poses, similarly to the dataset constructed in Peng et al. (2020), but with the addition of random $SO(3)$ rotations on each object. We call this dataset the *Seismic dataset*.

In figure 5a we show a quantitative comparison between our method (TF-Onet [E:0-2,D:0-2]) and previous methods that lack either the local shape modeling or the equivariant property. Figure 4b shows a qualitative comparison between the reconstructions achieved by these methods, while Figure 6a shows more examples of the reconstruction achieved by our model for scenes from the seismic dataset. Methods that perform global shape modeling (Onet, VNN) and are trained on single objects fail to generalize to scenes containing multiple objects. On the other hand, methods that perform local shape modeling but are not equivariant to SE(3) transforms (ConvOnet), can reconstruct objects in novel poses, but with reduced quality. Finally, the $SE(3)$ equivariant GraphOnet, that uses per-point features with long range dependencies, cannot generalize well on novel scenes when it is only trained on single objects. Our method benefits from both equivariance and local shape modeling, and is able to generalize to novel scenes achieving quality similar to that on single object reconstruction. In Figure 6b, we also show examples of reconstructions of realistic scenes captured from the Matterport3D dataset (Chang et al., 2017). These scenes contain between 6000 to 13000 points. While our method has only been trained with aligned single objects from ShapeNet represented as point clouds of 300 points, it successfully reconstructs complicated realistic scenes containing multiple objects in arbitrary positions and poses, even from unseen classes.

## 5 CONCLUSION

We proposed a novel SE(3)-equivariant coordinate-based model for shape reconstruction, consisting of two attention-based networks. By incorporating equivariance and local shape modeling, our method leverages the compositional structure of objects and scenes. These two properties allow our model to train on single-aligned objects and reconstruct novel objects and scenes. We evaluate our method against state-of-the-art $SO(3)$-equivariant and non-equivariant methods trained with augmentations and compare favorably in the single shape reconstruction category. Additionally, using our model trained on single aligned objects, we show that it can reconstruct novel scenes with quality similar to single object reconstruction, a task where other methods fail.

## 6 ACKNOWLEDGEMENTS

We gratefully acknowledge financial support by the following grants: NSF FRR 2220868, NSF IIS-RI 2212433, NSF TRIPODS 1934960, NSF CPS 2038873, ARL DCIST CRA W911NF-17-2-0181, ARO MURI W911NF-20-1-0080, ONR N00014-17-1-2093, and ONR N00014-22-1-2677.

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

# A  APPENDIX

## A.1  EVALUATION METRICS

In table 1, we present the F-Score, Chamfer-L1 distance and Intersection over Union (IoU) of the reconstruction achieved by various models. For the F-Score and Chamfer-L1 distance, we compute the reconstructed mesh using the Marching Cubes algorithm (Lorensen & Cline, 1987). Then, we uniformly sample 100,000 points from the reconstructed mesh, and compare them with the "ground truth" points sampled similarly from the "ground truth" mesh. The standard deviation of the results for both metrics due to the randomness introduced by the sampling of the 100,000 points is smaller than $10^{-5}$. As defined in Tatarchenko et al. (2019), the F-Score here is the harmonic mean of the precision and the recall of the reconstruction. Precision is the percentage of reconstructed points that lie within a certain distance $\tau$ of the ground truth points, and recall is the percentage of ground truth points that lie within a distance $\tau$ of the reconstructed points. We compute the F-Score with $\tau$ equal to $1\%$ of the side length of the reconstructed volume, and with $\tau$ equal to $2\%$ of the side length of the reconstructed volume. For the IoU metric, we uniformly sample 100,000 points from the whole space, and compare the predicted occupancy to the ground truth occupancy. The IoU is then computed by taking the ratio of points that are occupied according to both the predicted and the ground truth occupancy to the points that are occupied according to either the predicted or the ground truth occupancy.

## A.2  MODEL ARCHITECTURE, TRAINING AND TESTING DETAILS

For the self-attention feature extractor network $\mathcal{E}$, we use ten multi-headed SE(3) attention layers, and for the cross-attention occupancy network $\mathcal{T}$ we use another two. All multi-headed SE(3) attention layers use eight heads. For each point $\vec{x}_i$ of the point cloud, we define the neighborhood $\mathcal{N}(\vec{x}_i)$ as its $k$ nearest neighbors, where $k$ is chosen to be $5\%$ of the size of the point cloud (e.g., for 300 points, $k$=15). Each feature map in the layers of $\mathcal{E}, \mathcal{T}$ uses irreducibles up to type-2.

We train our model on the ShapeNet (Chang et al., 2015) subset constructed in Choy et al. (2016). We use the Adam (Kingma & Ba, 2015) optimizer with learning rate that starts at $2 \cdot 10^{-4}$ and linearly decreases to reach the value of $10^{-5}$. We train for 200,000 iterations using a batch size of 64. During training, we take as input a point cloud of 300 points and as ground truth the occupancy value of 2048 points that are sampled uniformly inside a box that bounds the object. As a training loss, we use the binary cross-entropy loss between the predicted and the ground truth occupancy of the 2048 randomly sampled points.

During inference, recalling that the output of our model for one query point is in the range of $[0, 1]$, we classify a query point as occupied if the value of the learned occupancy function is above $0.2$. After we query the occupancy of points throughout the space, we reconstruct the object's mesh using the Marching Cubes algorithm (Lorensen & Cline, 1987).

## A.3  EXTRACTING DIFFERENT TYPES OF REPRESENTATIONS AS FEATURES

An important quality of our method is its ability to use different types of representations as its intermediate features. These different types determine the way that the features transform as the input rotates. These transformation laws affect the shape properties that each type of features can encode. For example type-0 features can encode properties of the shape that are invariant to the rotation of the shape, type-1 features can encode properties that must rotate as 3d vectors (e.g. the normal vectors at the surface) and type-2 features can encode properties that rotate as symmetric matrices (e.g. the inertial matrix of the shape).

A type $l$ feature consists of a $2l + 1$ dimensional vector that rotates according to the corresponding Wigner-D matrix $D_l : SO(3) \rightarrow \mathbb{R}^{(2l+1)\times(2l+1)}$. In figure 7 we visualize the different types of features extracted by our encoder for different cases of inputs. The difference between the type of features can be clearly observed in the case of the symmetric sphere where features of higher type $l$ correspond to higher frequencies on the sphere.

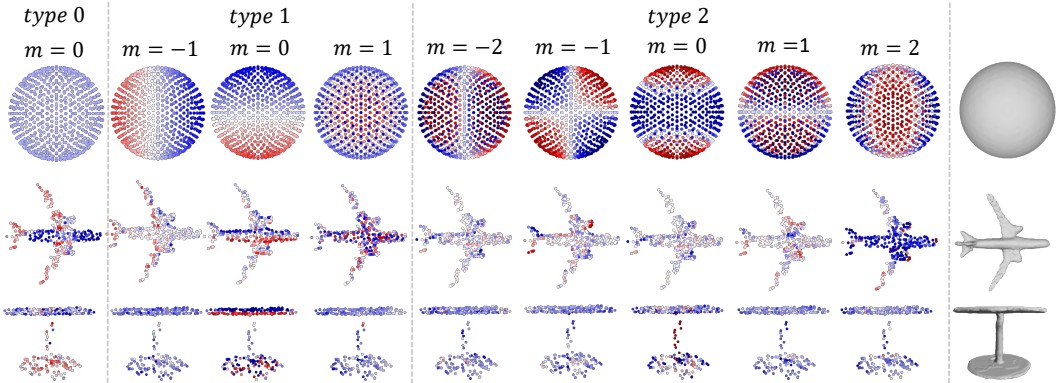

Figure 7: Visualization of each dimension of a type-$l$ feature extracted by our TF-Onet encoder (with $l = 0, 1, 2$). A type-$l$ feature corresponds to a $2l + 1$ dimensional vector where each dimension is denoted with index $m \in \{-l, \ldots, l\}$. The last right column shows the final reconstruction achieved by our method for the corresponding input.

## A.4 RECONSTRUCTIONS OF SCANNED SCENES

We further investigate the generalization of our method to novel scenes by reconstructing real objects of different domains and levels of clutter (Fig. 8). The input is a sparse, unoriented point cloud scanned from scenes consisting of an arbitrary number of randomly placed objects. The model is agnostic to the number of objects in the scene and is trained only on synthetic single objects. The model outputs high-quality reconstructions even in difficult settings with objects very close to each other. We observe small artifacts in very cluttered scenes, localized in regions of incidence between objects (e.g. mugs in line 4 of Fig. 8).

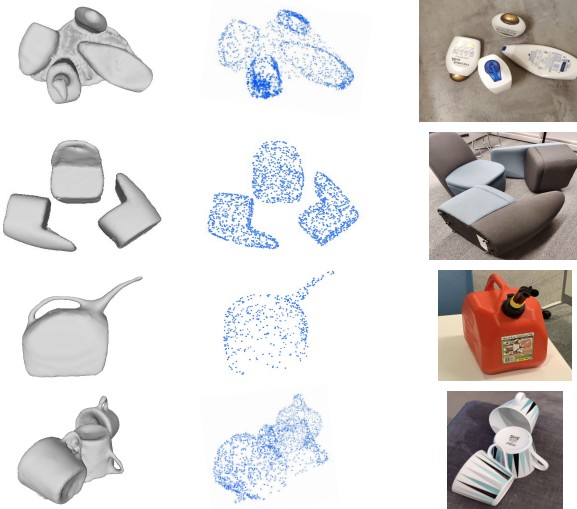

Figure 8: (Left): Reconstructions of real scans of randomly placed objects produced by our model which is trained only on single synthetic objects. (Middle): Input scene-level point cloud. (Right): RGB image of the scene.

## A.5 ROBUSTNESS TO CLUTTER

We evaluate the model on the *Seismic dataset* (Sec. 4.2) which contains scenes with varying levels of clutter. In Fig. 9 we measure the Intersection over Union (IoU) of the reconstruction versus the minimum distance between any two objects in the scene (normalized with respect to the room size).

The minimum distances range from around $6\%$ to $20\%$ of the room. The performance of the model is relatively stable across all settings with a maximum performance decrease of $0.025$ in IoU.

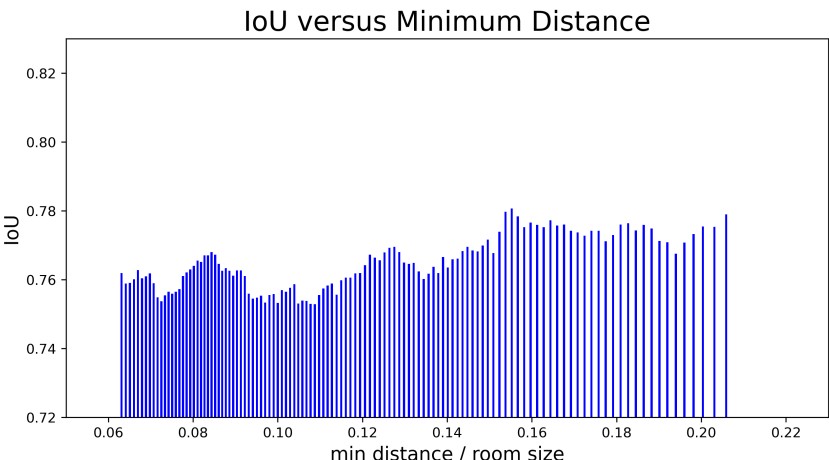

Figure 9: Intersection over Union (IoU) of the reconstructions from the *Seismic dataset* (Sec. 4.2) with respect to the minimum distance of any two objects in the scene.

### A.6    LIMITATIONS AND FUTURE WORK

A possible limitation of this work is the inherent memory overhead of the attention modules used in the network. In our setting, this overhead is lowered due to the locality of the attention operation but it can be further minimized with additional optimization of the attention modules. The optimization of the attention modules is an active research direction that can also benefit our method. An additional possible direction for future work is to extend our method to the problem of scene completion from partial observations. In this task, the model is required to hallucinate and reconstruct unobserved areas of a scene and thus requires a combination of local and global information about the scene configuration. While a significant methodological extension is required for our method to tackle such a task we believe that the core principles of equivariance via higher-order representations and local shape modeling can provide useful tools.

### A.7    WEIGHT PARAMETRIZATION

Here, we present the parametrization of the weights of the queries, keys, and values that appear as the solutions of Equation 7 (and written analytically in Equation 8). We focus on the specific case where both the input and output feature maps are $\rho$-fields consisting of up to type-2 irreducibles. Then, each feature in the feature maps can be decomposed into

$$f_{\text{in}} = \bigoplus_{k=0}^{2} \bigoplus_{m_k=1}^{M_k} f_{in}^{k,m_k}, f_{\text{out}} = \bigoplus_{k'=0}^{2} \bigoplus_{n_{k'}=1}^{N_{k'}} f_{\text{out}}^{k',n_{k'}}$$

For clarity, we start with multiplicities 1 i.e., $M_k = N_{k'} = 1$ (and omit the index of the layer). Then we extend to the general case.

The matrix $W_Q$ appearing in the query tokens $Q(\vec{x}_i, f_{\text{in}}) = W_Q f_{\text{in}}$ has the form:

$$
\begin{pmatrix} f^{0,1}_{\text{out}} \\ f^{1,1}_{\text{out}} \\ f^{2,1}_{\text{out}} \end{pmatrix} = \underbrace{\left( \begin{array}{c|c|c} w^{0,0} & 0 & 0 \\ \hline 0 & w^{1,1} I_3 & 0 \\ \hline 0 & 0 & w^{2,2} I_5 \end{array} \right)}_{W_Q} \begin{pmatrix} f^{0,1}_{\text{in}} \\ f^{1,1}_{\text{in}} \\ f^{2,1}_{\text{in}} \end{pmatrix}
$$

Similarly, we present the form of the function of matrices $W_K$, that appear in the key tokens as $K(\vec{x}_j, \vec{x}_i, f_{in}) = W_K(\vec{x}_j - \vec{x}_i) f_{in}$. To simplify the notation we use $\vec{x}$ instead of $\vec{x}_j - \vec{x}_i$ and $\hat{x} = \vec{x}/\|\vec{x}\|$. To write $W_K$ in matrix form we first need to define:

$$
\Phi^{k',k}_{l-u}(\|\vec{x}\|) \otimes C^{k',k}_{l-u}(\hat{x}) := \sum_{i=l}^{u} \underbrace{\phi^{k',k}_i(\|\vec{x}\|)}_{\text{learned}} \underbrace{C^{k',k}_i(\hat{x})}_{\text{fixed}}
$$

The functions $\phi^{k',k}_i : \mathbb{R}^+ \to \mathbb{R}$ are parametrized by MLPs. $C^{k',k}_i(\hat{x}) \in \mathbb{R}^{(2k'+1) \times (2k+1)}$ are fixed matrices defined in Section 8. Given the definition above we can write $W_K$ as:

$$
\underbrace{\left( \begin{array}{c|c|c} \Phi^{0,0}_{0-0}(\|\vec{x}\|) \otimes C^{0,0}_{0-0}(\hat{x}) & \Phi^{0,1}_{1-1}(\|\vec{x}\|) \otimes C^{0,1}_{1-1}(\hat{x}) & \Phi^{0,2}_{2-2}(\|\vec{x}\|) \otimes C^{0,2}_{2-2}(\hat{x}) \\ \hline \Phi^{1,0}_{1-1}(\|\vec{x}\|) \otimes C^{1,0}_{1-1}(\hat{x}) & \Phi^{1,1}_{0-2}(\|\vec{x}\|) \otimes C^{1,1}_{0-2}(\hat{x}) & \Phi^{1,2}_{1-3}(\|\vec{x}\|) \otimes C^{1,2}_{1-3}(\hat{x}) \\ \hline \Phi^{2,0}_{2-2}(\|\vec{x}\|) \otimes C^{2,0}_{2-2}(\hat{x}) & \Phi^{2,1}_{1-3}(\|\vec{x}\|) \otimes C^{2,1}_{1-3}(\hat{x}) & \Phi^{2,2}_{0-4}(\|\vec{x}\|) \otimes C^{2,2}_{0-4}(\hat{x}) \end{array} \right)}_{W_K(\vec{x})} \begin{pmatrix} f^{0,1}_{\text{in}} \\ f^{1,1}_{\text{in}} \\ f^{2,1}_{\text{in}} \end{pmatrix}
$$

The form of the matrices $W_V$ appearing in the value tokens as $V(\vec{x}_j, \vec{x}_i, f_{\text{in}}) = W_V(\vec{x}_j - \vec{x}_i) f_{\text{in}}$ is similar to $W_K$ above.

In the general case, if the input representation contains $M_k$ copies of type-k irreducibles we need to stack the corresponding blocks of $W_Q, W_K$, $M_k$ times in the column dimension. Similarly, if the output representation contains $N'_k$ copies of type-$k'$ irreducibles we need to stack the corresponding blocks $N_{k'}$ times in the row dimension. Specifically,

- for $W_Q$ and $k = k'$:

$$
\left[ w^{k,k} I_{2k+1} \right] \mapsto \begin{bmatrix} w^{k,k}_{(1,1)} I_{2k+1} & \cdots & w^{k,k}_{(1,M_k)} I_{2k+1} \\ \vdots & \ddots & \vdots \\ w^{k,k}_{(N_{k'},1)} I_{2k+1} & \cdots & w^{k,k}_{(N_{k'},M_k)} I_{2k+1} \end{bmatrix}
$$

- for $W_K$ and $l = |k' - k|, u = |k' + k|$:

$$
\left[ \Phi^{k',k}_{l-u} \otimes C^{k',k}_{l-u} \right] \mapsto \begin{bmatrix} [\Phi^{k',k}_{l-u}]_{(1,1)} \otimes C^{k',k}_{l-u} & \cdots & [\Phi^{k',k}_{l-u}]_{(1,M_k)} \otimes C^{k',k}_{l-u} \\ \vdots & \ddots & \vdots \\ [\Phi^{k',k}_{l-u}]_{(N_{k'},1)} \otimes C^{k',k}_{l-u} & \cdots & [\Phi^{k',k}_{l-u}]_{(N_{k'},M_k)} \otimes C^{k',k}_{l-u} \end{bmatrix}
$$

## A.8 PRELIMINARIES

We recall some basic notions from group and representation theory, see e.g., Fulton & Harris (2013). A group $(G, \cdot)$ is a set $G$ together with a binary operator "$\cdot$": $G \times G \to G$ that satisfies the following axioms:

- **Associativity:** $g \cdot (h \cdot f) = (g \cdot h) \cdot f$ for all $g, h, f \in G$
- **Identity:** there exists an element $e \in G$ such that $e \cdot g = g \cdot e = g$
- **Inverse:** for all $g \in G$, there exists $g^{-1} \in G$ such that $g^{-1} \cdot g = g \cdot g^{-1} = e$.

Given $G$, we say that each group element $g \in G$ acts on the space $X$ via an **action** $L_g : X \to X$ if $L_g$ satisfy the following two properties:

- If $e$ is the identity element of $G$ then $L_e[x] = x$ for all $x \in G$;
- $L_g \circ L_h = L_{g \cdot h}$ for all $g, h \in G$.

If for any $x, y \in X$ there exists $g \in G$ such that $L_g[x] = y$, then we call $X$ a homogeneous space for the group $G$.

When $X = V$ is a vector space, we can define the group action using a **linear group representation** $(V, \rho)$, where $\rho : G \to GL(V)$ is a map from group $G$ to the general linear group $GL(V)$. This means that using the linear operator $\rho(g)$ on $V$, we can define the group action $L_g$ on the vector space $V$ as $L_g[x] = \rho(g)x$ for all $x \in V$, $g \in G$. Then $(V, \rho)$ is a linear group representation if $\rho$ is a group homomorphism, i.e., $\rho(g \cdot h) = \rho(g)\rho(h)$ for all $g, h \in G$, and $\rho(e) = I_V$ is the identity operator over $V$. To simplify the notation, when the vector space $V$ that the group acts on is easily inferred from the context, we will use $\rho$ to denote the representation $(V, \rho)$.

Given a set of actions $L_g : X \to X$ for $g \in G$, and a set of actions $T_g : Y \to Y$ for $g \in G$, we say that a map $f : X \to Y$ is $(G, L, T)$-equivariant if for every $g \in G$:

$$T_g[f(x)] = f(L_g[x]) \text{ for all } g \in G, x \in X.$$

If $f$ is linear and equivariant (with respect to $(G, L, T)$), then it is called an intertwiner (with respect to $(G, L, T)$).

If there exists a subspace $W \subset V$ such that for all $g \in G$ and $w \in W$, we have that $\rho(g)w \in W$, then $W$ is a $G$-invariant subspace of $V$, and $(W, \rho)$ is a subrepresentation of $(V, \rho)$. All representations $(V, \rho)$ have at least two subrepresentations: $(0, \rho)$ and $(V, \rho)$. If a representation has no other subrepresentations, then it is called **irreducible**. Otherwise it is called reducible. A fundamental result is the following (Fulton & Harris, 2013).

### A.8.1 SCHUR'S LEMMA

Let $(V, \rho_V)$, $(W, \rho_W)$ be irreducible representations of $G$ acting on $V$ and $W$, respectively.

- If $V$ and $W$ are not isomorphic, then there are no nontrivial intertwiners between them.
- If $V = W$ are finite-dimensional vector spaces over $\mathbb{C}$, and if $\rho_V = \rho_W$, then all intertwiners are scalar multiples of the identity.

Now we list a number of fundamental examples, see e.g., Fulton & Harris (2013); Hall (2003).

### A.8.2 THE TRANSLATION GROUP $(\mathbb{R}^3, +)$:

$\mathbb{R}^3$ equipped with the addition operator "+" is a group that is isomorphic to the group of translations in the 3D space.

### A.8.3 THE SPECIAL ORTHOGONAL GROUP $SO(3)$:

$SO(3)$ is the group of $3 \times 3$ orthogonal matrices with determinant $+1$ equipped with multiplication; and is isomorphic to the group of all 3D rotations about the origin. $SO(3)$ is a compact group and as a result of the Peter-Weyl theorem, its linear representations can be decomposed into a direct sum of finite-dimensional, unitary, irreducible representations. Specifically a linear representation of $SO(3)$ decomposes as:

$$\rho(g) = Q^T \left[ \bigoplus_{J \geq 0} D_J(g) \right] Q \text{ for all } g \in G,$$

where $Q$ is a change of basis matrix and $J = 0, 1, \ldots D_J$ are $(2J + 1) \times (2J + 1)$ matrices known as the Wigner D-matrices. The Wigner D-matrices are the irreducible representations of SO(3). In the context of the features of a neural net, the representations (viewed as vectors) that transform according to $D_J$ are called type-$J$ vectors.

### A.8.4 THE SPECIAL EUCLIDEAN GROUP SE(3):

SE(3) is the group of proper rigid transformations of 3D space, and is isomorphic to the semidirect product $(\mathbb{R}^3, +) \rtimes \mathrm{SO}(3)$. An element of SE(3) can be represented as $(T, R)$ where $T$ is an element of the group of translations and $R$ is an element of the group SO(3) of 3D rotations. For two elements $(T_1, R_1), (T_2, R_2) \in \mathrm{SE}(3)$, the group law is defined as:

$$(T_1, R_1) \cdot (T_2, R_2) = (T_1 + R_1 T_2, R_1 R_2).$$

Since every point in $\mathbb{R}^3$ can be transformed into any other point with a proper rigid transformation, we have that $\mathbb{R}^3$ is a homogeneous space for SE(3).

In addition to the action of SE(3) on vectors in $\mathbb{R}^3$, we can also define the action of SE(3) on functions $f : \mathbb{R}^3 \to \mathbb{R}^M$, for any given integer $M > 0$. This action is called the induced representation $\pi = \mathrm{Ind}_{\mathrm{SO}(3)}^{\mathrm{SE}(3)} \rho$ of SE(3). It acts on $f$ as follows:

$$[\pi((T, R))f](x) = \rho(R)f(R^{-1}(x - T)),$$

where $(T, R) \in \mathrm{SE}(3)$, $x \in \mathbb{R}^3$ and $\rho$ is a representation of $SO(3)$. Especially in the context of neural nets where functions are viewed as feature fields, a function that transforms according to $\pi = \mathrm{Ind}_{\mathrm{SO}(3)}^{\mathrm{SE}(3)} \rho$ is called a $\rho$-field, and if $\rho$ corresponds to the $l$-th irreducible representation of SO(3), it is also called a field of type-$l$.

### A.9 ARCHITECTURE DETAILS

A feature-augmented point cloud $P = (X, F)$ where $X := [\vec{x}_1 \cdots \vec{x}_N] \in \mathbb{R}^{3 \times N}$ and $F := [f_1 \cdots f_N] \in \mathbb{R}^{3 \times M}$ for some $M \in \mathbb{N}$ can be associated with a 3d field $f : \mathbb{R}^3 \to \mathbb{R}^M$ of finite support by writing $f(x) = \sum_{i=1}^{N} f_i \delta(\vec{x} - \vec{x}_i)$. This association will be useful to define the group action on the feature-augmented point cloud and will unify the inputs and outputs of $\mathcal{T}, \mathcal{E}$ in the sense that both of them process fields in $\mathbb{R}^3$ and output fields in $\mathbb{R}^3$. We will interchangeably use $(X, F)$ and $f$ for the point cloud in the next sections. We will call $f$ the *point cloud function* when the distinction is not clear.

The main module in our architecture is an SE(3)-equivariant attention block. Each block consists of a multi-head SE(3)-equivariant attention module followed by a skip connection and an equivariant layer normalization step.

### A.9.1 MULTI-HEAD SE(3)-EQUIVARIANT ATTENTION MODULE:

This module consists of multiple heads that implement either self-attention (input features for keys, values and queries are the same) or cross-attention (input features for keys and values are the same and different from the inputs for the queries). We will first describe the self-attention variant of the module and then describe the changes that are required for the cross-attention version.

The self-attention SE(3)-equivariant module takes as input a function $f$ (describing the point cloud as discussed above) defined by $f(\vec{x}) = \sum_{i=1}^{N} f_i \delta(\vec{x} - \vec{x}_i)$. Each one of the $f_i$ vectors can be decomposed into irreducible representations of SO(3) appearing with different multiplicities. This means that a single vector $f_i$ can be decomposed as $f_i = \bigoplus_{l \geq 0} \bigoplus_{m_l \geq 0} f_{i_{l,m_l}}$, where $f_{i_{l,m_l}}$ corresponds to the $m_l$-th multiplicity of the irreducible component of type-$l$. Since we perform self-attention, the keys, values and queries are computed using the same input function $f$. Specifically, for pairs of key and query points $(\vec{x}_j, \vec{x}_i)$, we compute the keys $K(\vec{x}_j, \vec{x}_i, f_j) = W_K(\vec{x}_j - \vec{x}_i)f_j$, the queries $Q(\vec{x}_i, f_i) = W_Q f_i$, and the values $V(\vec{x}_j, \vec{x}_i, f_j) = W_V(\vec{x}_j - \vec{x}_i)f_j$. To ensure equivariance, $W_K$, $W_Q$, $W_V$ must satisfy the conditions described in Section 3.3.

Suppose that for the computed key and query features, the $l$-th irreducible appears with multiplicity $M_l$, and for the computed value features the $k$-th irreducible appears with multiplicity $N_k$. For each

irreducible, we split its multiplicities across the different heads of the attention block. Assuming that we have $H$ heads in total, each head $h$ receives keys $K^{(h)}(\vec{x}_j, \vec{x}_i, f_j)$ and queries $Q^{(h)}(\vec{x}_i, f_i)$ containing irreducibles with multiplicities $M_l/H$ and receives values $V^{(h)}(\vec{x}_j, \vec{x}_i, f_k)$ containing irreducibles appearing with multiplicities $N_k/H$. After this split, the output of the self-attention for each head can be computed as:

$$\text{SA}^{(h)}[X, f](\vec{x}_i) = \sum_{\vec{x}_j \in \mathcal{N}(\vec{x}_i)} \alpha_X \left( Q^{(h)}(\vec{x}_i, f_i), K^{(h)}(\vec{x}_j, \vec{x}_i, f_j) \right) V^{(h)}(\vec{x}_j, \vec{x}_i, f_j),$$

where

$$\alpha_X \left( Q(\vec{x}_i, f_i), K(\vec{x}_j, \vec{x}_i, f_j) \right) = \frac{\exp \left[ (Q(\vec{x}_i, f_i))^T K(\vec{x}_j, \vec{x}_i, f_j) \right]}{\sum_{\vec{x}_j \in \mathcal{N}(\vec{x}_i)} \exp \left[ (Q(\vec{x}_i, f_i))^T K(\vec{x}_j, \vec{x}_i, f_j) \right]}.$$

Similar to the keys, values and queries, the output can have irreducibles and multiplicities that differ from the input and decompose as:

$$\text{SA}^{(h)}[X, f] = \bigoplus_k \bigoplus_{n_k} \text{SA}^{(h)}[X, f]_{k, n_k},$$

where $\text{SA}^{(h)}[X, f]_{k, n_k}$ is the $n_k$-th multiplicity of the $k$-th irreducible.

After the application of the self-attention layer, we concatenate the output of all the heads to create $\text{SA}[X, f] = \bigoplus_h \text{SA}^{(h)}[X, f]$. Then we pass the concatenated output through a linear SE(3)-equivariant layer to take $\text{SA}^{\text{out}}[X, f] = W_P \text{SA}[X, f]$. Since this linear layer also needs to be equivariant, it must follow the same constraints as the query matrix $W_Q$. By Schur's lemma, it follows that $W_P$ can only mix feature vectors that correspond to the same irreducibles.

For implementing cross-attention, we use the same process as with self-attention, with the only difference that the inputs for the queries are different from the inputs for the keys and the values. As a result, the output of the cross-attention for a single head $h$ is computed as:

$$\text{CA}^{(h)}[X, f](f_q, \vec{q}) = \sum_{\vec{x}_j \in \mathcal{N}_X(\vec{q})} \alpha_X \left( Q^{(h)}(\vec{q}, f_q), K^{(h)}(\vec{x}_j, \vec{q}, f_j) \right) V^{(h)}(\vec{x}_j, \vec{q}, f_j).$$

### A.9.2 SKIP CONNECTION:

The skip connection concatenates the output features of the multi-headed SE(3)-equivariant module with the features of the input query. To respect the type of each feature, this concatenation must happen between features corresponding to the same irreducible.

Suppose that $f^{\text{out}}$ is the output of the multi-head attention, decomposing into irreducibles and their multiplicities as $f^{\text{out}} = \bigoplus_k \bigoplus_{n_k} f^{\text{out}}_{k, n_k}$. Similarly suppose that $f_q$ is the input query, decomposing into irreducibles and their multiplicities as $f_q = \bigoplus_l \bigoplus_{m_l} f_{q, (l, m_l)}$. The application of the skip connection gives an output

$$f^{\text{skip}} = \bigoplus_k \left[ \left( \bigoplus_{n_k} f^{\text{out}}_{k, n_k} \right) \oplus \left( \bigoplus_{m_k} f_{\vec{q}, (k, m_k)} \right) \right].$$

To keep the output at the same types and multiplicities as specified by the user (i.e., to be a $\rho_{out}$-field) we only concatenate multiplicities from the input that exist in the output. Also, after the skip connection we apply an equivariant linear layer between irreducibles of the same type to project the features to the correct dimensions.

### A.9.3 EQUIVARIANT LAYER NORM:

Suppose that $\mathbf{f}_{l,m}$ denotes the $m$-th type-$l$ irreducible of the input vector $\mathbf{f}$. As proposed in Fuchs et al. (2020), for each vector $\mathbf{f}_{l,m}$, we apply layer normalization and a nonlinearity on the norm of $\mathbf{f}_{l,m}$, leaving its direction unchanged. Thus, the equivariant normalization layer can be written as:

$$\text{EqLayerNorm}(f)_{l,m} = \text{ReLU} \left( \text{LayerNorm} \left( \bigoplus_m \|\mathbf{f}_{l,m}\| \right) \right)_m \frac{\mathbf{f}_{l,m}}{\|\mathbf{f}_{l,m}\|},$$

where LayerNorm corresponds to layer normalization, proposed in Ba et al. (2016).

We use the SE(3)-equivariant attention block to construct both the network $\mathcal{E}$ that assigns to each point in the point cloud a learned feature, and the network $\mathcal{T}$, that outputs the occupancy value of a query point in space. The network $\mathcal{E}$ consists of ten SE(3)-equivariant self-attention blocks, and $\mathcal{T}$ consists of two SE(3)-equivariant cross-attention blocks. Figure 2 shows a diagram of this architecture.

In both networks, we use blocks with eight heads and intermediate representations that contain features up to type-2. Additionally, for each type we set the multiplicity of the corresponding irreducible to 32. Finally, for the computation of the local neighborhood $\mathcal{N}(\vec{x})$ around each point $\vec{x}$, we use the $k$ nearest neighbors, where $k$ is chosen to be 5% of the size of the point cloud (e.g., for 300 points, $k$=15).

Although it is possible for $\mathcal{T}$ to output a single scalar that corresponds to the occupancy value at a queried point, we observe in the experiments an increase in performance when $\mathcal{T}$ outputs 32 scalar values that we then pass through a simple MLP to get the final occupancy value.

## A.10 PROOFS ON EQUIVARIANCE FROM SEC.3.3

**1. Input-output equivariance (Eq. 3):** We will formalize the equivariance constraints in the language of group theory. We have a map $\hat{o}$ that takes as input a point cloud $X \in \mathbb{R}^{3 \times N}$ and outputs an occupancy field $\hat{o}(X) : \mathbb{R}^3 \to \mathbb{R}$. First we need to define the actions of SE(3) on the input point cloud and the occupancy map which we denote by $\mathcal{L}_{\text{in}}, \mathcal{L}_{\text{out}}$ respectively. Those have the following form for $(T, R) \in SE(3)$:

$$\mathcal{L}_{\text{in},(T,R)}[X] = RX + \oplus_N T \tag{9}$$

$$[\mathcal{L}_{\text{out},(T,R)}\hat{o}(X)](\vec{q}) = [\hat{o}(X)](R^{-1}(\vec{q} - T)), \tag{10}$$

where $\oplus_N$ on vectors denotes concatenation column-wise. The first equation describes $N$ standard representations of SE(3) and the second the induced representation of SE(3) via SO(3) with $\rho(R) = I$, i.e., the map $\hat{o}$ is a scalar (or type-0) field.

For completeness we show that $\mathcal{L}_{\text{in}}$ indeed describes an action of SE(3) on $X$. Letting $(I, \oplus_N 0)$ be the identity element of SE(3), we can check that

$$\mathcal{L}_{\text{in},(I,\oplus_N 0)}[X] = IX + \oplus_N 0 = X.$$

Also, for any $(T_1, R_1), (T_2, R_2)$ from SE(3), we can check that

$$\begin{aligned}
\mathcal{L}_{\text{in},(T_2,R_2)}[\mathcal{L}_{(T_1,R_1)}[X]] &= R_2(R_1 X + \oplus_N T_1) + \oplus_N T_2 = \\
&= R_2 R_1 X + R_2(\oplus_N T_1) + \oplus_N T_2 \\
&= R_2 R_1 X + \oplus_N (R_2 T_1 + T_2) \\
&= \mathcal{L}_{\text{in}, R_2 T_1 + T_2, R_2 R_1}[X] \\
&= \mathcal{L}_{\text{in},[(T_2,R_2)\cdot(T_1,R_1)]}[X].
\end{aligned}$$

Using the vector space isomorphism associating $X \in \mathbb{R}^{3 \times N}$ with the unrolled vector $\text{vec}(X) \in \mathbb{R}^{3N}$, we can view this action as the direct sum of $N$ standard representations. Moreover, $\mathcal{L}_{\text{out}}$ is also an action, and in fact an *induced representation of SE(3) via SO(3) with* $\rho(R) = I, R \in SO(3)$, according to the definition A.8.4.

Now that we have the input and output actions, we can define the equivariance constraint for our problem. Informally, a simultaneous roto-translation of the point cloud and the query results in an invariant prediction. Formally,

$$\hat{o}(\mathcal{L}_{\text{in},(T,R)}[X]) = \mathcal{L}_{\text{out},(T,R)}\hat{o}(X) \iff \tag{11}$$

$$[\hat{o}(\mathcal{L}_{\text{in},(T,R)}[X])](\vec{q}) = [\mathcal{L}_{\text{out},(T,R)}\hat{o}(X)](\vec{q}), \quad \forall \vec{q} \in \mathbb{R}^3 \iff \tag{12}$$

$$[\hat{o}(RX + \oplus_N T)](\vec{q}) = [\hat{o}(X)](R^{-1}(\vec{q} - T)), \quad \forall \vec{q} \in \mathbb{R}^3 \iff \tag{13}$$

$$[\hat{o}(RX + \oplus_N T)](R\vec{q} + T) = [\hat{o}(X)](\vec{q}), \quad \forall \vec{q} \in \mathbb{R}^3. \tag{14}$$

Now we recall that we have parametrized $\hat{o}$ as a composition of a feature extractor $\mathcal{E}$ and an occupancy network $\mathcal{T}$, i.e., $[\hat{o}(X)](\vec{q}) = [\mathcal{T}(\mathcal{E}(X))](\vec{q})$. Thus we arrive to Eq. 3:

$$[\mathcal{T}(\mathcal{E}(RX + \oplus_N T))](R\vec{q} + T) = [\mathcal{T}(\mathcal{E}(X))](\vec{q}). \tag{15}$$

**2. Per-layer equivariance (Eq. 4, 5):** Next, we need to prove that the per-layer equivariance constraints in Eq. 4, 5 are sufficient to satisfy the input-output equivariance constraint from Eq. 3, provided that $\rho_{\mathcal{T}}^{L'+1}(R) = I, R \in SO(3)$. We recall that the forms of the feature extractor $\mathcal{E}$ and the occupancy network $\mathcal{T}$ are:

$$\mathcal{E}[X] = \mathcal{E}^L \cdots \circ \mathcal{E}^2 \circ \mathcal{E}^1 \circ \tilde{\mathcal{E}}^0[X] \tag{16}$$

$$\tilde{\mathcal{E}}^0[X] = (X, S(X)) \tag{17}$$

$$\mathcal{T}[X, F](\vec{q}) = \mathcal{T}^{L'} \cdots \circ \mathcal{T}^2 \circ \mathcal{T}^1 \circ \tilde{\mathcal{T}}^0[X, F](\vec{q}) \tag{18}$$

$$\tilde{\mathcal{T}}^0[X, F](\vec{q}) = (\vec{q}, S'(X, \vec{q})). \tag{19}$$

All $\mathcal{E}^1, \mathcal{E}^2, \cdots, \mathcal{E}^L$ take as input and produce as output a feature field on the point cloud and satisfy the constraints in Eq. 4. Similarly, all $\mathcal{T}^1, \mathcal{T}^2, \cdots, \mathcal{T}^{L'}$ take as input a feature field on the point cloud and a feature field in $\mathbb{R}^3$ and pass the point cloud feature field unaltered to the next layer. They transform the feature field in $\mathbb{R}^3$ and satisfy the constraints in Eq. 5. Thus, by composing these constraints in Eq. 4.5 we immediately find:

$$\mathcal{E}^L \cdots \circ \mathcal{E}^2 \circ \mathcal{E}^1(RX + \oplus_N T, \rho_{\mathcal{E}}^1(R)F^1) = (RX + \oplus_N T, \rho_{\mathcal{E}}^{L+1}(R)F^{L+1}), \tag{20}$$

$$\mathcal{T}^{L'} \cdots \circ \mathcal{T}^1[RX + \oplus_N T, \rho_{\mathcal{E}}^{L+1}(R)F^{L+1}](R\vec{q} + T, \rho_{\mathcal{T}}^1(R)f_q^1) = (R\vec{q} + T, \rho_{\mathcal{T}}^{L'+1}(R)f_q^{L'+1}),$$

where $\rho_{\mathcal{T}}^{L'+1}(R) = I$ as discussed. The equations above show that if every layer transforms an input $\rho$-field to an output $\rho$-field, then the composition also transforms an input $\rho$-field to an output $\rho$-field. Thus, it remains to show that the fixed layers $\tilde{\mathcal{E}}^0, \tilde{\mathcal{T}}^0$, namely $(X, S(X))$ and $(\vec{q}, S'(X, \vec{q}))$ respectively are $\rho$-fields as well. Then, the equations in Eq. 20 would not only apply from the first layer to the output, but from the input to the output. In other words, we need to check that the features produced by $S, S'$ do not translate when the point cloud translates but they do rotate (under suitable representations) when the point cloud rotates.

**2.1. Input point cloud field and input query field are type-1 fields**: For the $i$-th point at position $\vec{x}_i$, we construct a feature $f_i$ as input to the first self-attention layer of $\mathcal{E}$. We select the feature $f_i$ as the relative position of the $i$-th point, $\vec{x}_i$, to the centroid of a neighborhood constructed from its $k$ nearest neighbors in the point cloud in Euclidean norm, i.e.,

$$f_i := S(X)_i = \vec{x}_i - \frac{1}{|\mathcal{N}_i^k(X)|} \sum_{j \in \mathcal{N}_i^k(X)} \vec{x}_j, \qquad X = \oplus_{i=1}^N [\vec{x}_i],$$

where

$$\mathcal{N}_i^k(X) = \{j \in [N] \mid \|\vec{x}_i - \vec{x}_j\|_2 \leq \|\vec{x}_i - \vec{x}_k^{(i)}\|_2\}$$

is the neighborhood of the $i$-th point in the point cloud. Also, $(\vec{x}_j^{(i)})_{j \in [N]}$ is a sorting of the points in the point cloud in increasing Euclidean distance to the $i$-th point, i.e, $\|\vec{x}_i - \vec{x}_1^{(i)}\|_2 \leq \cdots \leq \|\vec{x}_i - \vec{x}_N^{(i)}\|_2$. In case of ties, we assign numbers randomly. Due to the use of " $\leq$ " (instead of the strict inequality symbol " $<$ "), in the definition of the neighborhoods, if there are ties for $\vec{x}_k^{(i)}$, we include all tied points in the neighborhood.

Now we study how the features $f_i$ transform when the points in the point cloud transform via $\mathcal{L}_{\text{in},(T,R)}$, discussed in the first step. Since

$$\mathcal{L}_{\text{in},(T,R)}[X] = \oplus_{i=1}^N (\mathcal{L}_{\text{in},(T,R)}[X]_i) = \oplus_{i=1}^N [R\vec{x}_i + T],$$

we find:

$$S(\mathcal{L}_{\text{in},(T,R)}[X])_i = \mathcal{L}_{\text{in},(T,R)}[X]_i - \frac{1}{|\mathcal{N}_i^k(\mathcal{L}_{\text{in},(T,R)}[X])|} \sum_{j \in \mathcal{N}_i^k(\mathcal{L}_{\text{in},(T,R)}[X])} \mathcal{L}_{\text{in},(T,R)}[X]_j$$

$$= (R\vec{x}_i + T) - \frac{1}{|\mathcal{N}_i^k(X)|} \sum_{j \in \mathcal{N}_i^k(X)} (R\vec{x}_j + T),$$

where we used the claim, proved below, that

$$\mathcal{N}_i^k(\mathcal{L}_{\mathrm{in},(T,R)}[X]) = \mathcal{N}_i^k(X). \tag{21}$$

Thus, $S(\mathcal{L}_{\mathrm{in},(T,R)}[X])_i$ further equals:

$$R\left(\vec{x}_i - \frac{1}{|\mathcal{N}_i^k(X)|}\sum_{j\in\mathcal{N}_i^k(X)}\vec{x}_j\right) = RS(X), \text{ for all } (T,R) \in \mathrm{SE}(3).$$

We now prove 21. When all $\vec{x}_i$ are mapped as $\vec{x}_i \xrightarrow{\mathcal{L}_{\mathrm{in},(T,R)}} R\vec{x}_i + T$, the Euclidean distance between any two points is preserved, i.e., $\|(R\vec{x}_i + T) - (R\vec{x}_j + T)\|_2 = \|\vec{x}_i - \vec{x}_j\|_2$. Thus, if before the transformation $\|\vec{x}_i - \vec{x}_k^{(i)}\|_2 = d_k$, then after the transformation we also have $\|\mathcal{L}_{\mathrm{in},(T,R)}[X]_i - \mathcal{L}_{\mathrm{in},(T,R)}[X]_k^{(i)}\|_2 = d_k$. This is because all nearest neighbors preserve their distances, and thus sorting returns the same indices up to random tie breaking. Thus,

$$\begin{aligned}
j \in \mathcal{N}_i^k(X) &\iff \|\vec{x}_i - \vec{x}_j\|_2 \le d_k \\
&\iff \|(R\vec{x}_i + T) - (R\vec{x}_j + T)\|_2 \le d_k \\
&\iff \|\mathcal{L}_{\mathrm{in},(T,R)}[X]_i - \mathcal{L}_{\mathrm{in},(T,R)}[X]_j\|_2 \le d_k \\
&\iff \|\mathcal{L}_{\mathrm{in},(T,R)}[X]_i - \mathcal{L}_{\mathrm{in},(T,R)}[X]_j\|_2 \le \|\mathcal{L}_{\mathrm{in},(T,R)}[X]_i - \mathcal{L}_{\mathrm{in},(T,R)}[X]_k^{(i)}\|_2 \\
&\iff j \in \mathcal{N}_i^k(\mathcal{L}_{\mathrm{in},(T,R)}[X]).
\end{aligned}$$

The first and fourth equivalence hold because we include *all* tied neighbors in the neighborhood. Thus, the neighborhood is defined by the *distance* $d_k$, and not by the identity of the $k$-neighbors.

Thus $f_i = S(X)_i$ transforms according to the standard representation of $\mathrm{SO}(3)$ when each point in the point cloud transforms according to the standard representation of $\mathrm{SE}(3)$. If we view the features as a function on the point cloud extended to the homogeneous space $\mathbb{R}^3 \cong \mathrm{SE}(3)/\mathrm{SO}(3)$, i.e., $f(x) = \sum_{i=1}^N f_i \delta(x - x_i)$, then this function transforms according to the *induced representation* as:

$$\mathcal{L}_{(T,R)}^{(ind)}[f](\vec{x}) = Rf(R^{-1}(\vec{x} - T)) = \sum_{i=1}^N (R\mathbf{f}_i)\delta((R^{-1}(\vec{x} - T) - \vec{x}_i)).$$

Recall that functions transforming according to the above law are called type-1 fields. We will keep the name, but use a matrix notation instead of the Dirac notation. By concatenating the features column-wise, we find the map, described in the main text as $S$, i.e., $S(RX + \oplus_N T) = \oplus_{i=1}^N Rf_i = RS(X)$.

Now we turn to the second network, $\mathcal{T}$. For each point $\vec{q} \in \mathbb{R}^3$ whose occupancy value we wish to find, we first construct a feature $f_q^1 := S'(X, \vec{q})$ and then use the pair $(\vec{q}, f_q^1)$ as the query to the first cross-attention layer of $\mathcal{T}$. We will show that, when the query $\vec{q}$ and the point cloud $X$ transform according to the standard representation of $\mathrm{SE}(3)$, this input feature $f_q^1$ also transforms according to the standard representation of $\mathrm{SO}(3)$.

We again construct the feature $f_q^1$ as the relative position between $\vec{q}$ and the centroid of its neighborhood $\mathcal{N}_X^Q(\vec{q})$. We consider $\mathcal{N}_X^Q(\vec{q})$ to be the same as the neighborhood of its closest—in Euclidean distance—point in the point cloud, and write (if the closest point is defined uniquely)

$$\mathcal{N}_X^Q(\vec{q}) := \mathcal{N}_{\arg\min_{i\in[N]}(\|\vec{q} - \vec{x}_i\|_2)}^k(X).$$

We discuss at the end of this step the case where the nearest neighbors are tied. At the moment, let the unique closest point be $c = \arg\min_{i\in[N]}(\|\vec{q} - \vec{x}_i\|_2)$. Then, the query feature becomes:

$$\begin{aligned}
f_q^1 := S'(X, \vec{q}) &= \vec{q} - \frac{1}{|\mathcal{N}_X^Q(\vec{q})|}\sum_{j\in\mathcal{N}_X(\vec{q})}\vec{x}_j \\
&= \vec{q} - \frac{1}{|\mathcal{N}_c^k(X)|}\sum_{j\in\mathcal{N}_c^k(X)}\vec{x}_j, \qquad X = \oplus_{i=1}^N[\vec{x}_i]. \tag{22}
\end{aligned}$$

The proof that

$$S'(RX + \oplus_N T, R\vec{q} + T) = RS'(X, \vec{q})$$

is the same as the one for $S$ before. Viewed again as a function defined on $\mathbb{R}^3$, the map $\vec{q} \mapsto S'(X, \vec{q})$ constitutes a type-1 field. The transformation law of this field is depicted in Fig. 3.

**2.2. Remark:** When there are ties, i.e., the set $\arg\min_{i\in[N]}(\|\vec{q} - \vec{x}_i\|_2)$ contains more than one point, we form all neighborhoods

$$\mathcal{N}_X^Q(\vec{q}) = \{\mathcal{N}_c^k(X) \mid c \in \arg\min_{i\in[N]} \|\vec{q} - \vec{x}_i\|_2\}.$$

Then, we consider a query token for each pair $(\vec{q}^{\mathbf{c}}, f_q^c)$, where $\vec{q}^{\mathbf{c}} = \vec{q}$ and $f_q^c$ are constructed as in 22, using the neighborhood $\mathcal{N}_c^k(X) \in \mathcal{N}_X^Q(\vec{q})$.

If $f_q$ transforms as a type-1 field, then, after the roto-translation of the point cloud and the query, we could identify the same $c$ as a closest neighbor. However, since those points are equivalent as nearest neighbors, we will process the whole set of fields independently, producing a set of fields in the output. A different order of selection of nearest neighbors after the roto-translation corresponds to a permutation of the set of the output fields. Since attention modules are permutation equivariant, this permutation propagates to the output.

We only need to discuss how to combine these outputs on the tokens $\vec{q}^{\mathbf{c}}$ that correspond to the same position $\vec{q}$ in the occupancy field. Since the output is a scalar field (as we prove next), we can take the maximum across the same channels to construct a new scalar field that is also invariant to any permutation. The idea is that taking the maximum corresponds to an "OR" operation, since both the usual non-linearities (such as the sigmoid) and the thresholding operations that follow are increasing functions of their inputs. Thus, when the network predicts that the position of the query is "occupied", even based on one neighborhood, the position is likely to be considered occupied.

**3. From per-layer constraints 4, 5 to constraints on the weights 6:** We focus now on each layer of $\mathcal{E}, \mathcal{T}$ separately. Now we need to prove that Eq. 6 provide sufficient conditions on the weights to satisfy the per-layer constraints 4, 5. Every layer is composed of a multi-headed attention layer, a skip connection and a normalization layer. We prove the result for the self-attention and cross-attention layers (focusing on one attention head for simplicity). It will be clear from the proof that a concatenation of the heads and a subsequent linear transformation mixing channels that correspond to the same irreducible preserves equivariance, as well as a skip connection that concatenates irreducibles of the same type.

We start with the self-attention layers in $\mathcal{E}$ described in A.9.1. We drop the layer index $l$ for clarity and denote the actions $\rho_\mathcal{E}^l, \rho_\mathcal{E}^{l+1}$ by $\rho_{\text{in}}, \rho_{\text{out}}$.

At the $i$-th token. we have

$$\text{SA}[X, F]_i = \sum_{j \in \mathcal{N}_i^k(X)} \alpha_X[Q(\vec{x}_i, f_i), K(\vec{x}_j, \vec{x}_i, f_j)] \underbrace{W_V(\vec{x}_j - \vec{x}_i) f_j}_{V(\vec{x}_j, \vec{x}_i, f_j)},$$

where the attention kernel $\alpha_X$ takes the form:

$$\begin{aligned}
\alpha_X[Q(\vec{x}_i, f_i), K(\vec{x}_j, \vec{x}_i, f_j)] &= \frac{\exp\left[(Q(\vec{x}_i, f_i))^T K(\vec{x}_j, \vec{x}_i, f_j)\right]}{\sum_{j \in \mathcal{N}_i^k(X)} \exp\left[(Q(\vec{x}_i, f_i))^T K(\vec{x}_j, \vec{x}_i, f_j)\right]} \\
&= \frac{\exp\left[(W_Q f_i)^T (W_K(\vec{x}_j - \vec{x}_i) f_j)\right]}{\sum_{j \in \mathcal{N}_i^k(X)} \exp\left[(W_Q f_i)^T (W_K(\vec{x}_j - \vec{x}_i) f_j)\right]}, \quad i \in [N].
\end{aligned}$$

Then, we are given

$$\begin{cases}
W_Q \rho_{\text{in}}(R) = \rho_{\text{out}}(R) W_Q \\
W_K(R(\vec{x}_j - \vec{x}_i)) \rho_{\text{in}}(R) = \rho_{\text{out}}(R) W_K(\vec{x}_j - \vec{x}_i) \\
W_V(R(\vec{x}_j - \vec{x}_i)) \rho_{\text{in}}(R) = \rho_{\text{out}}(R) W_V(\vec{x}_j - \vec{x}_i)
\end{cases} \tag{23}$$

and we need to prove:

$$\text{SA}(RX + \oplus_N T, \rho_{\text{in}}(R) F) = \rho_{\text{out}}(R) \text{SA}(X, F).$$

For the attention layer, we have for $X = \oplus_{i=1}^N [\vec{x}_i]$ and for each output token $i \in [N]$:

$$\text{SA}[RX + \oplus_N T, \rho_{\text{in}}(R)F]_i = \tag{24}$$

$$= \sum_{j \in \mathcal{N}_i^k(RX + \oplus_N T)} \alpha_{RX + \oplus_N T}[Q(R\vec{x}_i + T, \rho_{\text{in}}(R)f_i), K(R\vec{x}_j + T, R\vec{x}_i + T, \rho_{\text{in}}(R)f_j)] \cdot$$

$$W_V((R\vec{x}_j + T) - (R\vec{x}_i + T))\rho_{\text{in}}(R)f_j$$

$$= \sum_{j \in \mathcal{N}_i^k(X)} \alpha_{RX + \oplus_N T}[W_Q\rho_{\text{in}}(R)f_i, W_K(R(\vec{x}_j - \vec{x}_i))\rho_{\text{in}}(R)f_j]W_V(R(\vec{x}_j - \vec{x}_i))\rho_{\text{in}}(R)f_j,$$

where we used 21 for the invariant neighborhoods, $\mathcal{N}_i^k(RX + \oplus_N T) = \mathcal{N}_i^k(X)$. Now using the constraints for the matrices 23, we find for each term $j \in \mathcal{N}_i^k(X)$ that the individual terms in the sum above equal:

$$\alpha_{RX + \oplus_N T}[\rho_{\text{out}}(R)W_Q f_i, \rho_{\text{out}}(R)W_K(\vec{x}_j - \vec{x}_i)f_j]\rho_{\text{out}}(R)W_V(\vec{x}_j - \vec{x}_i)f_j$$

$$= \rho_{\text{out}}(R)\alpha_{RX + \oplus_N T}[\rho_{\text{out}}(R)W_Q f_i, \rho_{\text{out}}(R)W_K(\vec{x}_j - \vec{x}_i)f_j]W_V(\vec{x}_j - \vec{x}_i)f_j, \tag{25}$$

where in the second equation we used that the attention kernel gives a scalar output. Now, the attention kernel from the last equation transforms as

$$\alpha_{RX + \oplus_N T}[\rho_{\text{out}}(R)W_Q f_i, \rho_{\text{out}}(R)W_K(\vec{x}_j - \vec{x}_i)f_j] =$$

$$= \frac{\exp\left[(\rho_{\text{out}}(R)W_Q f_i)^T(\rho_{\text{out}}(R)W_K(\vec{x}_j - \vec{x}_i)f_j)\right]}{\sum_{j \in \mathcal{N}_i^k(RX + \oplus_N T)} \exp\left[(\rho_{\text{out}}(R)W_Q f_i)^T(\rho_{\text{out}}(R)W_K(\vec{x}_j - \vec{x}_i)f_j)\right]} =$$

$$= \frac{\exp\left[(W_Q f_i)^T(W_K(\vec{x}_j - \vec{x}_i)f_j)\right]}{\sum_{j \in \mathcal{N}_i^k(X)} \exp\left[(W_Q f_i)^T(W_K(\vec{x}_j - \vec{x}_i)f_j)\right]} = \alpha_X[Q(\vec{x}_i, f_i), K(\vec{x}_j, \vec{x}_i, f_j)], \tag{26}$$

where we used again the properties of the invariant neighborhoods and that $\rho_{\text{out}}$ is a unitary representation, i.e., $\rho_{\text{out}}(R)^T\rho_{\text{out}}(R) = I$ for all $R \in \text{SO}(3)$.

Using 25, 26, and 24, we find:

$$\text{SA}[RX + \oplus_N T, \rho_{\text{in}}(R)F]_i$$

$$= \rho_{\text{out}}(R) \sum_{j \in \mathcal{N}_i^k(X)} \alpha_X[Q(\vec{x}_i, f_i)K(\vec{x}_j, \vec{x}_i, f_j)]W_V(\vec{x}_j - \vec{x}_i)f_j$$

$$= \rho_{\text{out}}(R)\text{SA}[X, F]_i.$$

Thus each self-attention layer in $\mathcal{E}$ is equivariant. The proof for each cross-attention layer in $\mathcal{T}$ is similar. Again, note that if the closest neighbor of the query is not uniquely defined, then—as discussed above—we output the entire set of fields for every equivalent neighborhood. Then, the output is also a set of fields and a roto-translation of the input will only result in a permutation of these fields. Then, the "max" operation in the output will eliminate the permutation, making the output equivariant.

**4. From the constraints on the weights (Eq. 6) to their solutions (Eq. 7)**: Again, we focus on each layer separately as we did in the previous section. The goal is to solve Eq. 23.

By the Peter-Weyl theorem, $\rho_{\text{in}}$ decomposes into unitary, irreducible representations of SO(3), possibly with multiplicities. Thus,

$$\rho_{\text{in}}(R) = Q_{\text{in}}^T \left( \bigoplus_l \bigoplus_{m_l} \mathbf{D}_l(R) \right) Q_{\text{in}}, \ R \in \text{SO}(3),$$

where in our case $Q_{\text{in}} = I$, $\bigoplus$ for matrices denotes a concatenation along the diagonal and $l$ indexes the irreducible types and $m_l$ indexes the multiplicity of the $l$-th irreducible. Then, for each $\vec{x}$ in the point cloud (where we suppress the index $i$ for clarity), we have $\mathbf{f}^{\text{in}} = \bigoplus_l \bigoplus_{m_l} \mathbf{f}_{l,m_l}^{\text{in}}$. We also consider an output field transforming according to $\rho_{\text{out}}(R) = Q_{\text{out}}^T(\bigoplus_k \bigoplus_{n_k} \mathbf{D}_k(R))Q_{\text{out}}$—$Q_{\text{out}} = I$ in our case—that decomposes as $\mathbf{f}^{\text{out}} = \bigoplus_k \bigoplus_{n_k} \mathbf{f}_{k,n_k}^{\text{out}}$. Recall that each of the matrices $W_Q, W_K, W_V$ are of dimension $\sum_k n_k(2k+1) \times \sum_l m_l(2l+1)$.

1. For $W_Q$, we have for all $R \in \mathrm{SO}(3)$:

$$\rho_{\mathrm{out}}(R)W_Q = W_Q\rho_{\mathrm{in}}(R) \iff$$

$$[\bigoplus_{k,n_k} \mathbf{D}_k(R)]W_Q = W_Q[\bigoplus_{l,m_l} \mathbf{D}_l(R)] \iff$$

$$\mathbf{D}_k(R)W_Q = W_Q\mathbf{D}_l(R). \tag{27}$$

By Schur's Lemma (A.8.1), for each block $[W_Q]_{l,m_l}^{k,n_k}$ that multiplies $\mathbf{f}_{l,m_l}^{\mathrm{in}}$ to create $\mathbf{f}_{k,n_k}^{\mathrm{out}}$ (after adding all contributions), we have, for some constants $c_{l,m_l}^{k,n_k}$:

$$[W_Q]_{l,m_l}^{k,n_k} = \begin{cases} 0 & \text{if } l \neq k \\ c_{l,m_l}^{k,n_k} I_{2l+1} & \text{if } l = k. \end{cases} \tag{28}$$

Intuitively, the solution above says that the query cannot transform an irreducible type to a different irreducible type (e.g., a scalar to vector), but only mix channels that correspond to the same irreducible.

2. For $W_V$, the constraint is that for all $R \in \mathrm{SO}(3)$, the following set of equivalent statements holds:

$$W_V(R(\vec{x}_j - \vec{x}_i))\rho_{\mathrm{in}}(R) = \rho_{\mathrm{out}}(R)W_V(\vec{x}_j - \vec{x}_i) \iff$$

$$W_V(R(\vec{x}_j - \vec{x}_i))[\bigoplus_{l,m_l} \mathbf{D}_l(R)] = [\bigoplus_{k,n_k} \mathbf{D}_k(R)]W_V(\vec{x}_j - \vec{x}_i) \iff$$

$$[W_V(R_r(\vec{x}_j - \vec{x}_i))]_{l,m_l}^{k,n_k}\mathbf{D}_l(R) = \mathbf{D}_k(R)[W_V(\vec{x}_j - \vec{x}_i)]_{l,m_l}^{k,n_k}.$$

The solution, as discussed in the main text and solved in Weiler et al. (2018a); Thomas et al. (2018) is:

$$[W_V(\vec{x}_j - \vec{x}_i)]_{l,m_l}^{k,n_k} = \sum_{J=|l-k|}^{l+k} \phi_{V,(J,l,m_l)}^{(k,n_k)}(\|\vec{x}_j - \vec{x}_i\|; \theta)C_J^{k,l}\left(\frac{\vec{x}_j - \vec{x}_i}{\|\vec{x}_j - \vec{x}_i\|}\right),$$

where $C_J^{k,l}(\hat{x}) = \sum_{m=-J}^{J} Y_{Jm}(\hat{x})Q_{Jm}^{kl}$, $Q_{Jm}^{kl} \in \mathbb{R}^{(2k+1)\times(2l+1)}$ are the slices from the $Q^{kl}$ Clebsch-Gordan matrices, $Y_J : S^2 \to \mathbb{R}^{(2J+1)}$, is the $J$-th real spherical harmonic, $Y_{Jm}(\hat{x}) = [Y_J(\hat{x})]_m$ is its $m$-th coordinate, and $\hat{x} = \vec{x}/\|\vec{x}\|$.

3. For $W_K$, we have the same equation as for $W_V$ above:

$$W_K(R(\vec{x}_j - \vec{x}_i))\rho_{\mathrm{in}}(R) = \rho_{\mathrm{out}}(R)W_K(\vec{x}_j - \vec{x}_i),$$

but in addition we can constrain the blocks that transform to an irreducible that does not appear in the input to be zero without impacting the result. The reason is the form of $W_Q$ in Eq. 28. In particular, if $l', k$ are different irreducibles in the input and output respectively, then the blocks $[W_K]_{l',m_{l'}}^{k,n_k}$ that transform a type-$l'$ to a type-$k$ only contribute as terms to the total inner product in the attention kernel as follows.
For all $(l, m_l)$:

$$\langle [W_Q]_{l,m_l}^{k,n_k}D_l(R)f_l, [W_K(\vec{x}_j - \vec{x}_i)]_{l',m_{l'}}^{k,n_k}D_l(R)f_l\rangle$$

The above inner product is zero if $l \neq k$ due to Eq. 28. If $k \notin \mathcal{K}$, where $\mathcal{K}$ is the set of irreducibles appearing in the input (i.e. the range of $l$) then the inner product is zero for all $l, m_l$. So, we might as well choose:

$$[W_K(\vec{x}_j - \vec{x}_i)]_{l',m_{l'}}^{k,n_k} = 0, \text{ if } k \notin \mathcal{K}.$$

since parametrizing those blocks will not contribute to the result. For the rest of the blocks we have, similar as before,

$$[W_K(\vec{x}_j - \vec{x}_i)]_{l,m_l}^{k,n_k} = \sum_{J=|l-k|}^{l+k} \phi_{K,(J,l,m_l)}^{(k,n_k)}(\|\vec{x}_j - \vec{x}_i\|; \theta)C_J^{k,l}\left(\frac{\vec{x}_j - \vec{x}_i}{\|\vec{x}_j - \vec{x}_i\|}\right),$$

The parametrization of the weights of the cross-attention module $\mathcal{T}$ is similar to Eq. 7. The only difference is that we can further reduce the number of parameters without impacting the result by using only the type-1 features of the keys in the first layer, i.e., $[W_K]_{l,m_l}^{k,n_k} = 0$, for $k \neq 1$. This is due to the same equation involving the inner product between the key and the query we saw above, and the fact that the input query field is of type-1. In Fig. 3, we visualize the equivariance constraint on $\mathcal{T}, \mathcal{E}$ by using a commutative diagram.

**6. Additional Equivariant Layers**: Clearly, the concatenation of *multiple attention heads* with the same irreducible types, as well as *skip connection* layers as defined in section A.9, preserve equivariance. Since they add the multiplicities of each irreducible type independently, they only change the output representation by increasing the multiplicities of its irreducibles.

The subsequent mixing of channels of the same irreducible type by a linear map performed in the multi-headed attention module corresponds to the same operation that the query performs, thus it also preserves equivariance.

Finally, the equivariant layer-norm layer also operates on each irreducible type independently. Since the irreducibles are unitary transforms—i.e., $\|\mathbf{f}_{l,m_l}\|_2 = \|\mathbf{D}_l\mathbf{f}_{l,m_l}\|_2$ for all $\mathbf{f}_{l,m_l}$ and all $\mathbf{D}_l$—any non-linear operation on the norm of a *type-l* vector produces a type-0 vector. Since $\mathbf{f}_{l,m_l}/\|\mathbf{f}_{l,m_l}\|_2$ is again a *type-l* vector, the final result is a *type-l* vector.

**7. Attention as a set operation**: In addition to $\mathrm{SE}(3)$-equivariance, our model inherits the properties of the standard attention layers. Thus, it is equivariant to any permutation of the points in the point cloud, and to the order of the queries. Moreover, the number of output tokens can vary. We use this property for scene reconstruction, by reconstructing scenes of a variable number of points (and point clouds) even by training on single objects of a fixed number of points. The input (key-value) tokens can also change during inference, which we can use to adapt the neighborhoods dynamically during inference. This can counter that super-sampling a point cloud may reduce the receptive field.

**8. On independent $\mathrm{SO}(3)$ rotations.** Our local attention neighborhoods and equivariant reconstructions are particularly important properties when reconstructing scenes. Here the point clouds can be independently placed in arbitrary positions. It is natural to ask if we can connect the performance of our model in a particular scene to the performance in another scene, where the same point clouds have been independently roto-translated. For a single object, the true occupancy function should be the same under a simultaneous roto-translation of the point cloud and the query. Our equivariant pipeline respects this property, by outputting a scalar field. Thus, the performance of our model is consistent independently of the $\mathrm{SE}(3)$ transformations of a single point cloud.

Following a similar approach for scenes with multiple objects, one can associate an independent copy of $\mathrm{SO}(3)$ acting on each point cloud, i.e., an action $X^i \mapsto \mathcal{L}_{r_i}[X^i]$, for all objects $i \in I$. We can ask if there is a transformation of the query $\vec{q}$, that, after the rotation of the point clouds, results in the same occupancy value. Also, can this implemented without knowing the segmentation function that assigns the points to their point clouds? A reasonable first approach would be to transform the query $\vec{q}$ with the rotation matrix $R_i$ used to transform its closest neighbor.

Unfortunately, this transformation does not correspond to a group action. To understand this, consider two unit spheres at positions $(0,0,0)$ and $(1,0,0)$ in three-dimensional space, and the query point $\vec{q} = (-10,0,0)$. Then, consider the product group action $\mathcal{L}_1 = \mathcal{L}_{r_1,r_2} = \mathcal{L}_{(0,0,\pi),e}$ that rotates the first sphere around the $z$-axis by 180 degrees—$r_1 = (0,0,\pi)$—and fixes the second sphere—$r_1 = e$. Next, consider the reverse action $\mathcal{L}_2 = \mathcal{L}_{(0,0,-\pi),e}$. If we first multiply the corresponding group elements together, we find $\mathcal{L} = \mathcal{L}_1 \circ \mathcal{L}_2 = \mathcal{L}_{e,e} = I$. If we then find the nearest sphere and apply the identity action to the query, it will remain at $(-10,0,0)$. On the other hand, if we first find the nearest neighbor—in this case, the first sphere–and apply the corresponding action, then the query will rotate around the $z$-axis by 180-degrees, moving to $(10,0,0)$. If we then apply the second action by finding again the nearest neighbor—now the second sphere—the identity action will be applied and the query point stays at $(10,0,0)$. Thus, this transformation does not satisfy the "compatibility" property and thus it is not a group action.

Even though the conditions are not met for all query points in space, there are subsets of $\mathbb{R}^3$ that are closed under these transformations and for which these transformations indeed form group actions. We will consider equivariance only in those subsets. We construct them geometrically for two point

clouds for simplicity. Consider the point clouds $X_1, X_2$ rotating around points $c_1, c_2$, respectively. We construct the equivariant zone for the point cloud $X_1$. First take the point in $X_2$ that has the maximum distance (in Euclidean norm), say $d_2$, to its center of rotation $c_2$. Form the sphere $S_2$ around $X_2$, with a radius $d_2$ and center $c_2$. All of $X_2$ is contained in $S_2$. Connect the centers $c_1, c_2$, and denote the point of intersection of this segment with $S_2$ as $p$. Then, draw the segment between $p$ and $c_1$ and name the distance of $c_1$ to the middle point of this segment $D_1$. Every point in the sphere $S_1$ of radius $D_1$ and center $c_1$ is in the equivariant zone of $X_1$, called $Z_1$.

We prove that in the equivariant zone, our model is equivariant, for any independent rotation of the point clouds. By construction, every query $\vec{q}$ in the first equivariant zone $Z_1$ has its closest neighbor in the point cloud $X_1$. This holds for any rotation of any point cloud. Then, by definition, the action on the query is the same rotation $R_{r_1}$ that transformed the points in $X_1$. Since the point cloud and the query both rotate with the same transformation, the query neighborhood constructed within the cross-attention layers is invariant—leading to the same closest neighbor—as we proved for the single object case.

Now, suppose each point in $X_1$ has its $k$ nearest neighbors in $X_1$, for any rotation of the point clouds; this is reasonable for sufficiently dense or separated point clouds. Then these point cloud neighborhoods are also invariant after the individual rotations. Under these conditions, a direct product action of SO(3)-s on the point clouds with a simultaneous action of $R_1$ on the query in the first equivariant zone is viewed by the attention network as a simultaneous rotation of the point cloud $X_1$ and the query $\vec{q}$. This is because $\mathcal{N}(\vec{q})$ contains only points from $X_1$, so the attention module uses only key-value tokens from points in $X_1$. Further, the neighborhood of the query is invariant to a simultaneous rotation of $X_1$ and $\vec{q}$. Thus, as we proved in the single-object case, the occupancy value prediction for this transformed query is invariant to the transformation. Thus, for all queries in an equivariant zone, equivariance to the direct product action of SO(3) holds, i.e., for all $r_1, r_2, \cdots, r_I \in$ SO(3), and for all $\vec{q} \in Z_j$, with $j \in [I]$,

$$\mathcal{T}[(\mathcal{L}_{r_1}[X_1], \mathcal{L}_{r_2}[X_2], \cdots, \mathcal{L}_{r_I}[X_I]), R_j\vec{q}] = \mathcal{T}[(X_1, X_2, \cdots, X_I), \vec{q}].$$

