# OpenReview forum: "$\mathrm{SE}(3)$-Equivariant Attention Networks for Shape Reconstruction in Function Space"
_ICLR.cc/2023/Conference — ICLR 2023 poster_

### Official Review · Reviewer_YvKA · 2022-10-19

**Confidence:** 3
**Correctness:** 4
**Technical Novelty And Significance:** 3
**Empirical Novelty And Significance:** 3
**Recommendation:** 6

**Clarity, Quality, Novelty And Reproducibility:**

As mentioned above, the work presents clarity problems, which directly harm the potential reproducibility. However, the novelty looks significant, even if I cannot fully understand all the passages and check their correctness.

**Strength And Weaknesses:**

STRENGTH
===========
S1) GENERALIZATION: The method can generalize to complex scenes without seeing them at training time. I am not aware of other works with the same capability. This is a particularly compelling property.

S2) SIGNIFICANCE: The design of equivariant networks is an active research field which collects significant interest from Computer Vision, Graphics, and ML communities. The new technique proposed in the paper can open novel research directions with a consequent tangible impact.

WEAKNESSES
============
W1) CLARITY: I am not completely familiar with all the involved math, and not all passages are clear to me. Often, the maths heavily breaks the text (e.g., after Eq. 5), notation is pretty heavy (e.g., Eq. 6), and some background concepts are introduced during the method explanation interrupting the narration (e.g., after Eq. 6). In this sense, I highly recommend to revise the text, grouping symbols and concepts definition before method presentation, and well separating mathematical derivation from the textual narration.

W2) APPLICATIONS AND LIMITATIONS: The proposed method is tested only on the shape reconstruction task. Other reported networks have shown results on different applications and challenges. There is no mention of possible limitations, and I do not see methodological limits to applying the formulated attention also on other tasks (e.g., segmentation, classification, matching) with minor modification. Testing on different tasks (and also domains) is standard for novel feature extractors (e.g., Vector Neurons).


MINORS
=======
Q1) Is it possible to manipulate the Query Tokens to obtain a modification of the output shape?
Q2) How the model performs in scenes where target objects are not separated in space (i.e., they are close, as objects in clutter)?

**Summary Of The Paper:**

The paper proposes a novel architecture to obtain local shape modelling and SE(3) equivariance for the task of 3D shape reconstruction. These properties are obtained by generalizing the attention mechanism and imposing specific equivariance constraints on the layer weights. The method can be trained on single aligned objects, and at inference, time can naturally work both with single roto-translated objects and also with complex scenes.

**Summary Of The Review:**

Given that I am not high-confident about the math and parts of the explanation are confusing, I cannot assess the correctness of the method. I think the method requires substantial clarification before publication, and there are unjustified lacks in the experiments and no mention of limitations or future works. For this reason, I think the paper requires some work and cannot vote for acceptance yet.
Looking forward to the rebuttal; I will be happy to raise my score if it addresses my concerns.

---

> ### Author Response · Authors · 2022-11-15
> **Official Comment to Reviewer YvKA (1/2)**
>
> We appreciate the constructive criticism provided by the reviewer regarding our work and the suggestions for possible improvements. Next, we discuss in detail the revisions in our paper with additional experiments and changes in the manuscript. We hope these address any concerns and we are open to further suggestions.
>
> **Regarding the clarity of the presentation of our method:**
> We made revisions in Sections 3.2 and 3.3 which we believe will simplify the presentation of the method while at the same time retaining the precision of the mathematical  definitions and derivations. In particular,
> * In Section 3.3 we moved the discussion of the selection of irreducibles for shape reconstruction after the layer parametrization. This disentangles the practical from the theoretical part.
> * In Section 3.3 we simplify the notation by suppressing the point cloud $X$ from the output of the layers since it passes through unaltered.
> * We adapted the Introduction to highlight the main contributions of the method and the differences with other similar works.
> * In the Appendix we provide a visualization of the feature maps extracted from the encoder, grouped into different irreducibles, to further enhance intuition.
> * We motivated the use of the Peter-Weyl theorem for solving the equivariance constraints, and we simplify its presentation (Section 3.3).
>
> We are open to additional suggestions from the reviewer for improvements on the presentation of our method.
>
> **Regarding the application of our method in different point cloud tasks:**
> In this work, we propose a method for learning an equivariant neural field with a focus on surface reconstruction, following a large body of work [1,2,3,4] that also studies and provides solutions that are specific to this problem. Some of the design choices we made (e.g. the local shape modeling) are beneficial for surface reconstruction but may not be ideal for a general point cloud feature extractor.
>
> Specifically, leveraging local shape modeling allows our method to reconstruct objects out of the training distribution and even novel scenes. However, this might not be a suitable property for other standard point cloud tasks like part segmentation or object classification where the encoder needs to utilize more global information about the object.
>
> Nevertheless, we believe that some of the methods and concepts discussed in this work (e.g. achieving equivariance via higher-order representations or local attention modules) provide useful tools that can be utilized in multiple shape analysis tasks.
>
> $[1]$ Peng et al. Convolutional occupancy networks, ECCV 2020.
>
> $[2]$ Mescheder et al.
> Occupancy networks: Learning 3d reconstruction in function space, CVPR 2019.
>
> $[3]$ Chibane et al. Implicit Functions in Feature Space for 3D Shape Reconstruction and Completion, CVPR 2020.
>
> $[4]$ Chen et al. 3D Equivariant Graph Implicit Functions, ECCV 2022.
>
> **Regarding the possible manipulation of the Query Tokens to obtain a modification of the output shape:**
> Editing and manipulating learned neural fields is a very exciting and relatively unexplored research area. A compelling property of our method is that the features extracted by our equivariant encoder roto-translate according to known laws. Given a scene, we can apply these transformations directly to the features extracted from individual objects and use the local, equivariant decoder to produce a transformed neural field that corresponds to a different configuration of the scene.  These transformations are applied to the keys and values of the decoder while the query tokens stay fixed. We are open to any further suggestions on how to utilize our method for similar manipulation of neural fields.
>
> **Regarding the performance of our model in cluttered scenes:** We added in Appendix A.5 an additional ablation study showing how the distance between the objects in a scene affects the performance of our model. We observe that the performance of the model is relatively stable across different levels of clutter, with only a small performance decrease of around 3\% in IoU.
>
> Moreover, the robustness of the model to congested scenes can be observed in the additional experiments in Appendix A.4  where we reconstruct real scenes of manually scanned objects.

---

> > ### Author Response · Authors · 2022-11-15
> > **Official Comment to Reviewer YvKA (2/2)**
> >
> > **Regarding our experimental evaluations:**
> > We provide multiple experiments to justify the effectiveness of our method:
> > * Single object reconstruction on aligned and rotated objects following VNN. In this setting, we compare against state-of-the-art methods ONet, VNN, IFNet, NeuralPull, ConvOnet, and GraphOnet. Our equivariant method achieves consistent performance regardless of the rotation of the input and outperforms all other equivariant and non-equivariant methods.
> > * Quantitative and qualitative comparison on scene reconstruction from single object training. We show that the equivariance and local shape modeling of our method provides an advantage over previous works that lack some of these properties.
> > * An ablation study (Table 1) justifying the benefit of using higher-order representations (e.g. type-2) in the context of surface reconstruction.
> > * An ablation study on how the performance of our model is affected by different levels of noise and sparsity in the input (Figure 5b).
> > * An ablation study showing how the distance of objects in the scene affects the performance of our method ( Appendix A.5).
> > *  Additional examples of reconstructions of real scenes: Matterport3D (Fig 6b), manually scanned real objects (Appendix A.4). These results emphasize the generalization ability of our method to the reconstruction of novel scenes.
> >
> > **Regarding the limitation and future works:** We added a discussion in the Appendix on the limitations of our method and possible future directions.
> > * One possible limitation is the memory overhead due to the attention modules. While this is a common problem in such architectures in our case this problem is partially alleviated by the fact that the queries attend only a local neighborhood instead of all the keys. Reducing this overhead is an active research direction that can also benefit our method.
> > * Another direction for improvement and possible future work is to extend the method to the problem of scene completion from partial observations. This is a separate task from scene reconstruction in which the model needs to hallucinate unobserved areas of the scene.
> >
> > **More changes in the revised version not mentioned above:**
> > * Comparison with the concurrent work of Chen et al. 2022: 3D Equivariant Graph Implicit Functions, (GraphOnet) [Table 1] (as suggested by Reviewer dRvs)
> > * Visualization of the intermediate features extracted by our encoder  for different types of irreducible representations (Appendix A.3) (as suggested by Reviewer dRvs)

---

> > > ### Comment · Reviewer_YvKA · 2022-11-23
> > > **Reply about the paper**
> > >
> > > I would thank the Authors for their effort in replying to my concerns.
> > >
> > > Initially, my primary concerns were related to the paper presentation and the method's applicability (both shared by Reviewer dRvs). In the reviewed version of the paper, I see some efforts to improve the explanation, and I appreciated the discussion on limitations. I consider this positive, while for the camera ready, I keep suggesting to ease the reading by limiting the math breaks in the text paragraphs. Also, I appreciate the discussion on different applications, while many works test proposed novel feature extractors on other tasks or contexts (Vector Neurons, DiffusionNet, KPConv, DeltaConv). If changing the target application requires a major modification of the architecture, it is a drawback that should be mentioned.
> > >
> > > Other reviews question the novelty and request further experiments, and authors provided further reasonable evidence to support their contribution.
> > >
> > > In the end, I am not completely satisfied and keep my concerns, but I see the work's merits and strengths pointed out by other reviewers. For this reason, I raised my score to 6 while keeping my suggestion for the final version to work more on the text clarity and consider providing further experiments on other tasks to explore the potentiality of the proposed method fully.

---

> > > > ### Author Response · Authors · 2022-12-02
> > > > **Comment on Reviewer's Reply**
> > > >
> > > > We would like to thank the reviewer for the valuable comments and for updating the recommendation score. We believe that the feedback was constructive and helped us highlight the contributions of the paper. For the final version, we will keep working on separating the mathematical derivations from the discussion in order to improve conciseness further. We are open to specific suggestions on sections of the text that can be further improved.
> > > >
> > > > Additionally, we will add the references suggested by the reviewer and provide a more detailed discussion about how our method differs from these general feature extractors. Our method is designed specifically for the task of surface reconstruction, leveraging locality and equivariance via higher-order representations. While these properties are beneficial for reconstruction, as we showed in the experiments in Section 4 (ablations, comparisons; including general feature extractors like VNN), they may not be sufficient or necessary for other tasks (like shape completion, classification, and segmentation). We hope that our positive results on surface reconstruction will motivate further research that utilizes these properties for more shape analysis tasks.

---

### Official Review · Reviewer_dRvs · 2022-10-23

**Confidence:** 4
**Correctness:** 3
**Technical Novelty And Significance:** 3
**Empirical Novelty And Significance:** 2
**Recommendation:** 6

**Clarity, Quality, Novelty And Reproducibility:**

The paper is well written, sound and of high quality. It is probably not easily reproduced without code due to the more complex Wigner-D basis (but the authors provide code). There is some originality within this work by bringing the harmonic basis in a 3D point cloud architecture.

**Strength And Weaknesses:**

Strengths:
- Utilizing harmonic bands of degree > 1 makes sense to allow the network to model symmetries within the input. To my knowledge, it is first applied to 3D reconstruction in this work. Similar ideas have been seen in 2D before (harmonic networks).
- I like that the SE(3) equivariant formulations are well grounded in group theoretic considerations of irreducable representations.
- The paper is technically sound
- The paper is well written

Weaknesses:
- The novelty of the work is limited. There have been SE(3)-equivariant networks in a very similar setting before [1]. The notable addition to this work are the additional harmonic bands.
- The paper leaves out some of those work in its comparisons.
- In general, it is not evaluated if the additional harmonic bands (type-2) improve the results. The claim by the authors is not supported in experiments or theory. An ablation study, showing the benefits, is needed. Also, the a qualitative evaluation of these feature maps would be interesting. It might be that one can actually see different behaviour on symmetric objects.


**Summary Of The Paper:**

This work presents an SE(3)-equivariant transformer architecture for occupancy prediction from point clouds. The network uses two types of attention layers, one attention layer on edges in the point graph and one cross attention layer to aggregate information from local neighborhoods in the output layers.
Further, the network utilizes Wigner-D harmonic bands of degree 2, allowing the network to model certain symmetries in the data.
The network is evaluated on I/I, I/SO(3) and SO(3)/SO(3) setups on ShapeNet, a synthetic scene dataset based on ShapeNet and Matterport3D, a real scene dataset.

**Summary Of The Review:**

I lean towards rejecting this work, based on the above mentioned problems. I am not fully convinced by the experiments. There are many methods doing 3D reconstruction with neural networks now and I have seen much more detailed reconstructions [1, 2, 3] against which this method is not compared. The authors claim that their SE(3)-equivariant formulation is very beneficial for scene reconstruction but at the same time, they only show very few examples of 3D reconstructions on real datasets and do not provide a quantitative evaluation or ablation studies.

[1] Chen et al.: 3D Equivariant Graph Implicit Functions, ECCV 2022

[2] Jiang et al.: Local Implicit Grid Representations for 3D Scenes, CVPR 2020

[3] Chabra et al.: Deep Local Shapes, ECCV 2020

Minor notation problems:
- sometimes, x_i is used to describe the relative position (Fig2), sometimes to describe the absolute position (3.1)
- neighboring point pairs are sometimes denoted as x_i, x_j, and sometimes as x,y

---

> ### Author Response · Authors · 2022-11-15
> **Official Comment to Reviewer dRvs (1/2)**
>
> We would like to thank the reviewer for taking the time to review our paper.  We agree that formulating our method in the language of representation theory provides useful tools for surface reconstruction. Specifically, in our work, we incorporate those tools to design an expressive equivariant architecture via the usage of high-order representations.
>
> **Regarding the comment of the reviewer about the differences with the work of Chen et al. 2022 [1]:**
> The work of Chen et al.2022 (GraphOnet) is concurrent to this work, and the code was not publicly available at the time of submission. We contacted the authors and we added a comparison with their method in the revised version of the paper as suggested by the reviewer.
>
> We outperform GraphOnet both quantitatively and qualitatively in both of the following settings:
> * SO(3)-single object reconstruction from sparse point clouds (300 points) (Section 4.1, Table 1)
> * SE(3)-scene reconstruction when the models are trained on single objects (Section 4.2, Figure 5a)
>
> One of the main differences between our work and GraphOnet is our use of general/expressive linear layers grounded on representation theory. These layers allow us to use high-order representations (e.g. type-2) which as shown in the added ablation (Table 1) can benefit the reconstruction. Another difference from GraphOnet is that we use local attention modules to enhance the expressivity of the network since equivariant linear layers constrain the orientation parameters.
>
> In the revised version of the paper, we added in the Introduction an additional discussion about  the differences between our method and other equivariant works on shape reconstruction (VNN, GraphOnet)
>
>
>
> **Regarding the  evaluation of the claim that additional harmonic bands can improve the reconstruction results:**
> We provide an additional ablation study of the effectiveness of different representation types (Table 1). Specifically, we evaluate our model in three different configurations:
>
> * In the first one, both the encoder and the decoder use up to type-1 representations (E:0-1,D:0-1).
> * In the second one, the encoder uses up to type-2 representations while the decoder is constrained to  only up to type-1 representations  (E:0-2,D:0-1).
> * In the third one, we use up to type-2 representations both in the encoder and the decoder (E:0-2,D:0-2).
>
> We observe that our model achieves the best performance in the third setting when it uses up to type-2 representations both in the encoder and the decoder.   Additionally, our method outperforms both GraphOnet and VNN which are constrained to only use up to type-1 representations.
>
> **Regarding the comparison with the works of Jiang et al. 2020 [2], Chabra et al. 2020[3]:** None of the [2,3] operate on unoriented point clouds. They assume different inputs (oriented point clouds, SDF) which makes the reconstruction task different than ours. This is why the state-of-the-art methods that we compare against in this paper (especially the equivariant methods VNN, and GraphOnet) do not evaluate their method against [2,3].
> * About [2] (Local Implicit Grid Representations for 3D Scenes): This paper assumes oriented point clouds during inference (points plus normals). This makes the reconstruction problem significantly easier,  which justifies the visual differences in the reconstructions too. In the setting of [2], which learns a signed-distance field (SDF), the known normals provide information about the gradient of the learned  field close to the surface of the object. This additional information can simplify the problem.
>
>
>   Additionally, normals are not usually provided by standard sensors used in many robotics applications.
> * About [3] (Deep Local Shapes): This paper assumes SDF values sampled in space during inference. This is a different setting from ours, which is why this paper does not compare with unoriented point cloud reconstruction methods either.
> * We have acknowledged both of these works in the Related Work section.
>
> **Regarding the qualitative evaluation of the feature maps:**
>     We follow the suggestion of the reviewer and we added in Appendix A.3 a visualization of intermediate features extracted from our encoder for shapes with different symmetries. We group these features into their corresponding types and show how type-2 features differ in behavior compared to type-0, and type-1.

---

> > ### Author Response · Authors · 2022-11-15
> > **Official Comment to Reviewer dRvs (2/2)**
> >
> > **Regarding the reproducibility of the results due to the Wigner-D matrices:** Implementations for the Wigner-D matrices are provided in all standard equivariant neural network libraries (e.g. e3nn), which are publicly available.
> >
> > **Regarding the experimental results of our method on scenes:**
> >  * We have provided quantitative comparisons on the Seismic dataset, where we  outperform previous methods with a large performance gap (Figure 5a).
> >  This result establishes the ability of  our method to generalize to novel scenes when trained only on single objects.
> > * From all the methods we compare against, only ConvOnet shows examples of reconstructed scenes from Matterport3D. Similarly, we provide examples of reconstructions from real scenes from Matterport3D to emphasize the generalization ability of our model to novel object categories.
> > * We further support our claims on the generalization of the method to novel scene reconstruction even beyond standard experimental settings, by providing reconstructions from manually-captured scans of real scenes (Appendix A.4).
> >
> > **Regarding the notation:**
> > * In Fig.2, $\vec{x}_i$ is used as the absolute position of the point (while $f_i$ is the relative position to the centroid) as in (3.1).
> > * $\vec{x},\vec{y}$ were used as the generic arguments of the function while $\vec{x}_i,\vec{x}_j$ as the evaluation of the function. In the revision, we changed the notation to include only $\vec{x}_i,\vec{x}_j$ to avoid confusion.
> >
> > **More changes in the revised version not mentioned above:**
> >
> > * Ablation study showing the robustness of our method in cluttered scenes (Appendix A.5)(as suggested by Reviewer YvKA)
> > * Revisions in the text in Sections 3.2, 3.3 in order to improve the clarity of the presentation of the method. Furthermore, following the suggestion of Reviewer Ut5F we added in the Appendix an example with the solutions of our equivariant constraints for different types of irreducibles representations (Appendix A.7).
> >
> > We are open to further suggestions and feedback regarding our work. If the additional experiments and clarifications addressed your concerns  we would appreciate it if you re-evaluated your recommendation.

---

> > > ### Comment · Reviewer_dRvs · 2022-11-19
> > > **Thanks**
> > >
> > > I thank the authors for providing the additional comparisons with GraphONet and the ablation studies regarding harmonic bands. These were my main two concerns with this work.
> > > I also appreciate the addition of robustness experiments and limitations. In total, the presented method is sufficiently evaluated now in my opinion.
> > >
> > > Some remaining concerns are the complicated method (and therefore at parts hard to read paper as Reviewer YvKA pointed out). Also, I am still skeptical if this method can produce good reconstructions of larger, complete scenes. The shown examples mostly consist of a set of rotated objects and not full scenes with ground planes, walls, etc.
> > >
> > > I will raise my score to 6 to reflect the authors additions. I think this paper can be accepted for the contribution of using additional harmonic bands that are shown to clearly improve on previous work.

---

> > > > ### Author Response · Authors · 2022-12-02
> > > > **Comment on Reviewer's Answer**
> > > >
> > > >  We would like to thank the reviewer for providing a thorough review of our paper. We appreciate the suggestion to include the ablation experiments and the additional comparisons that we believe strengthen the claims of the paper. For the final version, we will keep working on the presentation of the method and we are open to additional suggestions that will underline the contributions of the paper even further.
> > > >
> > > >  We appreciate that the reviewer acknowledges the novelty of using higher-order representations for surface reconstruction and for reevaluating the recommendation score.

---

### Official Review · Reviewer_Ut5F · 2022-10-25

**Confidence:** 3
**Correctness:** 4
**Technical Novelty And Significance:** 3
**Empirical Novelty And Significance:** 4
**Recommendation:** 8

**Clarity, Quality, Novelty And Reproducibility:**

The manuscript is very dense and somewhat challenging to read, but given the deeply technical mathematical backing of the proposed method this is to be expected. I personally found it necessary to reference prior work (e.g. SE(3)-Transformers and Tensor Field Networks) to fully understand the approach. Again, this is not necessarily a bad thing, but the manuscript could perhaps suggest doing so. The appendices were useful but also quite extensive. Some more concrete example might be useful, e.g. showing the values of the Wigner-D matrices for a particular order J and / or showing the block sparsity structure of the weight matrices for a particular feature vector with a particular multiplicity of each order.

**Strength And Weaknesses:**

Ultimately, this manuscript presents a straightforward application of an existing technique (SE(3) Transformers) to an existing problem (occupancy modeling). The novelty is somewhat limited, but the work is timely and does contribute novel and valuable experiments to the literature. Occupancy modeling is a great application for SE(3)-equivariant networks, which remove the need for rotational data augmentation to handle arbitrary rotations. Even more compelling, when combined with local feature extraction, the proposed method can be applied to infer scenes comprising novel combinations of familiar shapes, which would otherwise require data augmentation that generates an immense number of combinations of shapes and poses. The experiments clearly demonstrate the value of the approach compared to a variety of baselines.

**Summary Of The Paper:**

This manuscript describes a novel method to reconstruct continuous surfaces of scenes or objects from sparse point clouds. The method applies recent advances in SE(3)-equivariant transformer architectures to this problem. The transformer uses an encoder-decoder architecture. The encoder is an SO(3)-equivariant transformer which computes a per-point feature vector based on a local neighborhood of points (the encoder is translation invariant because only relative positions of neighbors are used). Different parts of the feature vector can be interpreted as being scalar-valued (as in a traditional network), vector-valued (as in Vector Neurons), or tensor-valued for arbitrarily high-order tensors. Because of the geometric interpretability of these representations, an SE(3) transformation of the input points induces a corresponding transformation of the representations output by the encoder. The decoder then takes as input the point cloud, the features output by the encoder, and a query point anywhere in 3D, and outputs an occupancy value. This network then implicitly defines a continuous occupancy field. The decoder is also SE(3)-equivariant, such that passing a transformed point cloud to the network results in an occupancy field transformed by the same SE(3) transformation. The manuscript demonstrates how this can be used to infer shapes in novel poses that were not seen during training, which has implications for both training efficiency and composability of representations.

**Summary Of The Review:**

This paper is a good example of a step change in performance; the proposed networks process input point clouds in a fundamentally different way (i.e. they are SE(3)-equivariant) from prior works, and the result is a large gap in performance in scenarios where this is important (e.g. inference of shapes in a different pose than they appeared in training. The novelty may be limited, but the empirical contribution is nevertheless compelling.

---

> ### Author Response · Authors · 2022-11-15
> **Official Comment to Reviewer Ut5F**
>
> We appreciate the positive feedback and insightful comments regarding our work. We agree with the reviewer that occupancy modeling is a fitting application for SE(3) equivariance and combined with local shape modeling can constitute a strong prior for shape analysis tasks. We hope that the tools from representation theory exposed in this work will motivate further applications in 3d geometric tasks.
>
> **Regarding the density of the text:**
> We made revisions to the notation and the presentation of some aspects of the method in Sections 3.2, and 3.3 in order to improve their clarity. We also followed the suggestion of the reviewer and we added in Appendix A.7 an example of solutions for the equivariant constraint for different types of irreducible representations.
>
> **Other changes in the revised version:** Additional to the experiments found in the original paper on SO(3) shape reconstruction, SE(3) scene reconstruction and reconstruction of real scenes from Matterport3D, in the revised version we added the following:
> * Comparison with the concurrent work of Chen et al. 2022: 3D Equivariant Graph Implicit Functions, (GraphOnet) [Table 1] (as suggested by Reviewer dRvs)
> * Ablation study on  how different types of representations affects the performance of our method [Table 1] (as suggested by Reviewer dRvs)
> * Ablation study showing the robustness of our method in cluttered scenes (Appendix A.5) (as suggested by Reviewer YvKA)
> * Qualitative results from reconstructions of  manually scanned real scenes.
> * Visualization of the intermediate features extracted by our encoder  for different types of irreducible representations (Appendix A.3) (as suggested by Reviewer dRvs)

---

### Official Review · Reviewer_i4in · 2022-10-25

**Confidence:** 3
**Correctness:** 4
**Technical Novelty And Significance:** 3
**Empirical Novelty And Significance:** 3
**Recommendation:** 6

**Clarity, Quality, Novelty And Reproducibility:**

Both the theoretical formulation and experimental results are of high quality.

The paper appears to be novel, although the novelty relative to existing equivariant attention networks is not entirely clear.

The part on matrix groups and Peter-Weyl is a bit dense in terms of math notation. Some intuitive explanations can be helpful.

**Strength And Weaknesses:**

Strengths:

(1) Both the self-attention and cross attention models are well designed and prove to be effective.

(2) The design of SE(3) equivariant attention is based on rigorous group representation consideration.

(3) The experimental results on compositional generalization are impressive.

Weakness:

The difference and novelty relative to existing equivariant attention networks should be explained more carefully.

**Summary Of The Paper:**

This paper proposes SE(3)-equivariant coordinate-based model for shape reconstruction. The model consists of a self-attention model to encoder the points in the point cloud and a cross-attention model to output the occupancy of any query point. Both models are made SE(3) equivariant. The local attention and the SE(3) equivariance allows compositional generalization.


**Summary Of The Review:**

This paper makes a solid contribution to the problem of shape reconstruction from point cloud. The proposed self-attention and cross-attention modules are interesting and useful.

---

> ### Author Response · Authors · 2022-11-15
> **Official Comment to Reviewer i4in**
>
> We would like to thank the reviewer for taking the time to review our paper. We appreciate the positive comments on the formulation and experimental contributions of our method. We also believe that the concepts and tools developed in this paper will be beneficial to shape analysis tasks.
>
> **Regarding improvements in the clarity of our text:** We made revisions to the text in order to simplify the presentation of our method. These changes include the part of Section 3.3 where we describe the use of the Peter-Weyl theorem to solve the equivariance constraints. Additionally, to facilitate the comprehension of the reader we added in Appendix A.7 an example with solutions for these equivariance constraints.
>
> **Regarding the need for additional discussion of the differences between our method and other similar works on equivariance:** We added a more detailed discussion in the introduction of the revised paper. Specifically,
> * Our method utilizes the expressivity of the equivariant attention modules to extract local features that allow it to generalize to novel objects and scenes. On the other hand, previous methods on surface reconstruction either use only global features (VNN) or per-point features with long-range dependencies (GraphOnet). In the experimental section we provide an experiment on scene reconstruction showcasing how local shape modeling benefits  our method (Section 4.2, Figure 5a),4b) )
> * Additionally, our method is able to utilize higher-order (type-2 and above) representations and outperforms VNN and GraphOnet which are constrained to only type-0 (scalar) and type-1 (vector) representations due to the use of Vector Neurons. In the experimental section, we added an ablation on how these higher-order representations benefit the performance of our method (Table 1).
>
> **More changes in the revised version not mentioned above:** Additional to the experiments found in the original paper on SO(3) shape reconstruction, SE(3) scene reconstruction and reconstruction of real scenes from Matterport3D, in the revised version  of the paper we added the following:
> * Comparison with the concurrent work of Chen et al. 2022: 3D Equivariant Graph Implicit Functions, (GraphOnet) [Table 1] ( as suggested by Reviewer dRvs)
> * Ablation study showing the robustness of our method in cluttered scenes (Appendix A.5)(as suggested by Reviewer YvKA)
> * Qualitative results from reconstructions of  manually scanned real scenes (Appendix A.4).
> * Visualization of the intermediate features extracted by our encoder  for different types of irreducible representations (Appendix A.3) (as suggested by Reviewer dRvs)

---

### Decision · Program_Chairs · 2023-01-20

**Decision:**

Accept: poster

**Justification For Why Not Higher Score:**

While the results are impressive and the architecture well designed, the proposed approach dies depend on prior SE3 transformers work for the formulation and there is a limited technical innovation in that context.

**Justification For Why Not Lower Score:**

The proposed methods shows clear improvements over prior work and presents a well-designed architecture for a common task. All the reviewers are unanimous in recommending acceptance.

**Metareview: Summary, Strengths And Weaknesses:**

The paper introduces a principally designed SE3 equivariant network for predicting implicit shape representations from point cloud observations. The proposed network, building on earlier SE3-Transformers work, allows impressive performance on recovering dense reconstruction from a single point cloud, while also enabling zero-shot generalization to multi-object scenes. The reviewers unanimously recommend acceptance, and the AC concurs.

**Note From Pc:**

if the above contains the word "oral" or "spotlight" please see: "oral" presentation means -> notable-top-5% and "spotlight" means -> notable-top-25%. As stated in our emails, we are disassociating presentation type from AC recommendations

**Summary Of Ac-Reviewer Meeting:**

N/A